



# Reconstructing N2-fixing cyanobacterial blooms in the Baltic Sea beyond observations using 6- and 7-methylheptadecanes in sediments as specific biomarkers

Jérôme Kaiser[1], Norbert Wasmund[1], Mati Kahru[2], Anna K. Wittenborn[1], Regina Hansen[1], Katharina Häusler[1], Matthias Moros[1], Detlef Schulz-Bull[1], Helge W. Arz[1]

[1]Leibniz Institute for Baltic Sea Research (IOW), Seestrasse 15, 18119 Rostock-Warnemünde, Germany
[2]Scripps Institution of Oceanography, University of California San Diego, La Jolla, CA, 92093-0218, USA

*Correspondence to*: Jérôme Kaiser (jerome.kaiser@io-warnemuende.de)

**Abstract.** Summer cyanobacterial blooms represent a threat for the Baltic Sea ecosystem, causing deoxygenation of the bottom water and the spread of the so-called dead zones. The time history of the Baltic Sea cyanobacterial blooms is known from in situ and satellite observations since the early 1980s, but still not well understood. By comparing both weekly-resolved trap sediments and a well-dated sediment core from the Eastern Gotland Basin with monitoring and satellite cyanobacterial data of the last ca. 35 years, it is shown here that 6- and 7-methylheptadecane lipids (expressed as 6+7Me-$C_{17:0}$) are robust semi-quantitative biomarkers for diazotrophic cyanobacteria, and likely mainly for *Nodularia spumigena*. Using this organic proxy, it was thus possible to reconstruct the history of cyanobacterial blooms beyond the observational period with a resolution of 2-4 years since 1860. Cyanobacteria were constantly present, but in relatively low abundance until 1920, when they started to alternate between periods with high and low abundance. Interestingly, there seems to be no significant increase in cyanobacterial abundance in the 1950s, when eutrophication and deoxygenation of the Baltic Sea increased considerably. Decadal to multi-decadal fluctuations are likely rather related to variability in the Baltic Sea surface temperature and, ultimately, to the Atlantic Multidecadal Oscillation. A 7,000 years long 6+7Me-$C_{17:0}$ record from the Bothnian Sea also suggests a relationship with the mean summer temperature in the Baltic Sea region, but at a multi-centennial to multi-millennial timescale. The intensity of the cyanobacterial blooms in the Baltic Sea is thus likely mainly related to natural processes such as temperature variability, at least at a multi-decadal to multi-millennial timescale.

## 1 Introduction

Late summer (July-August) massive accumulations of $N_2$-fixing (diazotrophic) filamentous heterocystous cyanobacteria from the family Nostocaceae are a regular phenomenon in the surface layer of the Baltic Sea (Kahru et al., 1994; Kahru and Elmgren, 2014; Wasmund et al., 2018). These cyanobacterial blooms, that can cover an area of ca. 200 000 $km^2$ in the central Baltic Sea, are dominated primarily by the species *Nodularia spumigena*, but also *Aphanizomenon* sp., and, to a minor extent, *Dolichospermum* spp. and other species from the family Nostocaceae (Congestri et al., 2000; Stal et al., 2003;



Stal et al., 1999; Larsson et al., 2001; Wasmund et al., 2011, 2018; Hajdu et al., 2007; Hällfors, 2004; Kanoshina et al., 2003; Karjalainen et al., 2007; Sivonen et al., 2007; Celepli et al., 2017). These cyanobacteria aggregate near the surface during calm and warm weather (Kononen, 1992; Finni et al., 2001). If diazotrophic cyanobacteria occur in large blooms, they contribute to nitrogen-eutrophication, and massive export and decay of cyanobacterial biomass, and its associated $O_2$ consumption, and it causes the spread of bottom water hypoxia and anoxia (Zillen and Conley, 2010; Feistel et al., 2016).

Furthermore, *Nodularia spumigena* and *Dolichospermum* spp. have the potential to be toxic, whereas the toxicity of *Aphanizomenon* sp. has not been confirmed yet in the Baltic Sea (Wasmund, 2002; Luckas et al., 2005).

While cyanobacterial blooms have occurred in the Baltic Sea for thousands of years (Bianchi et al., 2000; Poutanen and Nikkilä, 2001; Jilbert et al., 2015; Funkey et al., 2014; Sollai et al., 2017; Szymczak-Żyła et al., 2019), it is often assumed that their frequency and intensity have increased due to anthropogenic impact. Long term monitoring programs indicate that

the frequency and abundance of diazotrophic cyanobacteria have increased since the early 1900s, and more significantly since the 1970s (Kahru et al., 1994), due to the massive anthropogenic loading of nutrients (in particular phosphorus) to the Baltic Sea (Finni et al., 2001; Poutanen and Nikkilä, 2001; Stal et al., 2003, Vahtera et al., 2007). As cyanobacteria grow better at higher temperature compared to pelagic microalgae, the on-going global warming may represent a catalyst for further expansion of cyanobacterial blooms in the Baltic Sea and their dominance in many other aquatic ecosystems (Pearl

and Huisman, 2008).

Biomarkers considered to be characteristic for cyanobacteria in modern and past ecosystems include some carotenoid pigments (zeaxanthin and echinenone; Lotocka, 1998; Bianchi et al., 2000; Poutanen and Nikkilä, 2001; Funkey et al., 2014; Jilbert et al., 2015; Szymczak-Żyła et al., 2019). However, zeaxanthin and echinenone are not entirely specific to cyanobacteria and are certainly not limited to nitrogen-fixing cyanobacteria (Bianchi, 2007; Sollai et al., 2017). Furthermore,

carotenoids are amongst the most unstable organic biomarkers because of their very labile conjugated system of double bonds (Britton et al., 2008; Fleischmann and Zorn, 2008). Some bacteriohopanepolyols (BHPs; Summons et al., 1999; Talbot et al., 2008), as well as heterocyst glycolipids (HGs) that have been reported exclusively from the heterocyst cell envelope of heterocystous cyanobacteria (Gambacorta et al., 1999; Bauersachs et al., 2009) are also used as cyanobacterial biomarkers in sediments (Gibson et al., 2008; Blumenberg et al., 2013; Sollai et al., 2017). Information on the diagenetic

stability of BHPs and HGs is currently only sparse, but they show potential to preserve well in the sedimentary archive, particularly under strongly reducing conditions. While BHPs specific for cyanobacteria were not found in Baltic Sea sediments (Blumenberg et al., 2013), HGs appear as robust markers for the study of diazotrophic cyanobacteria in Holocene Baltic Sea sediments (Sollai et al., 2017). Probably the first lipids to be suggested as exclusively produced by cyanobacteria (diazotrophic and non-diazotrophic), and thus having a strong potential as cyanobacterial biomarkers, are midchain branched

alkanes (Han and Calvin, 1969; Gelpi et al., 1970; Jüttner, 1991; Summons et al., 1996; Köster et al., 1999). Normal and mid-chain branched alkanes are amongst the most refractory lipids and do not experience major alterations during diagenesis (Peters et al., 2005). Given their exceptionally high preservation and the high specificity of mid-chain branched alkanes (Coates et al., 2014; Bauersachs et al., 2017), these components are considered best suited to examine the presence of





diazotrophic cyanobacteria in the modern and past Baltic Sea and possibly also other aquatic ecosystems. Furthermore,
compared to other analytical methods involving cost intensive instruments (Summons et al., 1999; Talbot et al., 2003, 2007;
Bauersachs et al., 2017), these compounds can be measured easily by gas chromatography - flame ionization detection (GC-FID).

Cyanobacteria possess the unique capacity to naturally produce hydrocarbons from fatty acids. Alkanes (e.g. heptadecane)
and branched alkanes (e.g. 7-methylheptadecane) are produced via a two steps conversion of fatty acids first to fatty
aldehydes and then alkanes that involves a fatty acyl ACP reductase (FAAR) and aldehyde deformylating oxygenase (ADO)
(Schirmer et al., 2010). 11-octadecanoic acid (vaccenic acid) is very likely the precursor to the methylheptadecanes (Han et
al., 1968; Fehler and Light, 1970; Coates et al., 2014). However, the mechanism for the introduction of branched groups
(methyl, ethyl, etc.) in cyanobacteria remains unknown (Liu et al., 2013). The physiological or ecological function of alkane
production in cyanobacteria is not known yet, but may be related to prevention of grazing, chemical signalling
(pheromones), prevention of desiccation, enhanced buoyancy, or membrane fluidity and stability (Coates et al., 2014).
Branched alkanes, most commonly as 7-methylheptadecane, were predominantly observed in particular clades including
heterocystous, ramified and some filamentous cyanobacteria, and rarely in unicellular cyanobacteria (Liu et al., 2013; Coates
et al., 2014). Recently, Bauersachs et al. (2017) examined the lipid profiles of heterocystous cyanobacteria strains belonging
to the genera *Dolichospermum*, *Aphanizomenon* and *Nodularia*, isolated from the Baltic Sea. Mid-chain branched alkanes
were not detected in any of the investigated *Dolichospermum* strains, although 7-methylheptadecane ($7Me\text{-}C_{17:0}$) and 8-
methylheptadecane ($8Me\text{-}C_{17:0}$) have been reported in a representative of this genus before (Coates et al., 2014; Fehler and
Light, 1970). In the *Aphanizomenon* strain, small quantities of monomethyl alkanes were also detected with $8Me\text{-}C_{17:0}$ being
the most dominant (7%) followed by $7Me\text{-}C_{17:0}$ (3%). The hydrocarbon profile of *Nodularia spumigena* included two
dimethyl heptadecanes, $7,11Me\text{-}C_{17:0}$ and $6,12Me\text{-}C_{17:0}$, in relative abundances of 5-28% and <5%, respectively. However,
the profile was dominated by $7Me\text{-}C_{17:0}$ and 6-methylheptadecane ($6Me\text{-}C_{17:0}$), which together accounted for 65% to 88% of
all hydrocarbons. As $6Me\text{-}C_{17:0}$ and $7Me\text{-}C_{17:0}$ were not detected in the other heterocystous cyanobacteria investigated (with
the exception of $7Me\text{-}C_{17:0}$ occurring in minor abundance in the *Aphanizomenon* strain), they can be considered as most
diagnostic for tracing planktonic cyanobacteria of the genus *Nodularia* in the Baltic Sea (Bauersachs et al., 2017). Indeed,
$7Me\text{-}C_{17:0}$ has been found in the surface waters of the central Baltic Sea (Landsort Deep) in summer (Berndmeyer et al.,
90    2014).

The aim of the present study is to evaluate if $6Me\text{-}C_{17:0}$ and $7Me\text{-}C_{17:0}$ represent specific biomarkers for the most abundant,
potentially toxic diazotrophic cyanobacteria *Nodularia spumigena* and, to a minor extent, *Aphanizomenon* spp., in Baltic Sea
sediments. For this purpose, sediment traps covering the period from May 2010 to January 2011 in the central Baltic Sea
(Eastern Gotland Basin, EGB) are investigated for the relative abundance of cyanobacteria and their related specific
biomarkers. Then, a well-dated short sediment core from nearly the same location (Fårö Deep) is analysed in order to
reconstruct the past occurrence of diazotrophic cyanobacterial bloom in the central Baltic Sea since 1860 with a sub-decadal



temporal resolution (2-4 years). Finally, a Holocene sediment core from the Bothnian Sea is analysed with a mean temporal resolution of 170 years.

## 2 Material and Methods

Sinking particles were collected from May 2010 to January 2011 in the Eastern Gotland Basin (EGB; Fig. 1; Table S1) at a water depth of 180 m with a classical cone-shaped automated Kiel sediment trap with a sampling area of $0.5 \ m^2$ (Zeitzschel et al., 1978). The collection intervals were set to 7 or 10 days. Buffered formalin (4%) was used as a fixative. After recovery, large zooplankton organisms were removed by sieving (400 μm gauze). Aliquots of the sediment trap material were qualitatively analysed for the main phytoplankton species (1: present; 2: abundant, 3: highly abundant) using an inverted

microscope (ZEISS Axiovert 200). The determination of the total flux of sedimentary matter were performed as described by Struck et al. (2004). The short sediment core MSM51-2/20 was retrieved with a device keeping the sediment-water interface undisturbed. The core was sampled every 0.5 to 1 cm and the sediment samples were frozen-dried and homogenized (n = 73). The long sediment core POS435/10 was sampled every 10-20 cm and the sediment samples were frozen-dried and homogenized (n = 45). The age model of core POS435/10 has been published in Häusler et al. (2017).

**2.1 Total organic carbon, XRF scanning, $^{137}$Cs, and PCBs analyses**

TOC was calculated by the subtraction of total inorganic carbon from total carbon analysed with an EA1110 CHN (CE-instruments) and a Multi EA 2000 CS (Analytik, Jena) elemental analyser (Leipe et al., 2011). Polychlorinated biphenyls (PCBs), $^{137}$Cs, and XRF data were used to elaborate the age model of core MSM51-2/20 following the approach developed in Moros et al. (2017). The activity of $^{137}$Cs radionuclide was measured by gamma spectrometry with a CANBERRA

BE3830 broad energy germanium detector (Moros et al., 2017). The analysis of PCBs was performed with gas chromatography coupled to mass spectrometry (Thermo Fisher Scientific Trace DSQ GC-MS) (Schulz-Bull et al., 1995; Moros et al., 2017). The sum content of the most abundant 23 PCB congeners was used here. An ITRAX X-ray fluorescence (XRF) scanner (Cox Analytical Systems) equipped with Cr-tube operated at 30 kV and 30 mA (exposure time of 15 seconds) was used to estimate the elemental composition of core MSM51-2/20 sediments with a 200 μm resolution. Mn, Fe

(expressed as count per second, cps), and the inc/coh ratio (e.g. Chawchai et al., 2015) were used here to help correlate sediment cores EMB1215/7 (Moros et al., 2017) and MSM51-2/20 (this study) together.

### 2.2 Age model

The age model of core MSM51-2/20 (42 cm-long; Fig. 2; Table S2) was obtained by correlating time markers defined in sediment core EMB1215/7 (39 cm-long), retrieved from about the same location (Fig. 1), and published in Moros et al.

(2017). Briefly, using an event stratigraphy approach Moros et al. (2017) have assigned dates to the sediment using (1) the early increase in Hg and $^{210}$Pb$_{unsupp.}$ that occurred around 1900, (2) the early increase in PCBs in 1935, (3) the beginning and





maximum in atom weapons tests in 1953 and 1963, respectively, as recorded by $^{241}$Am, (4) the $^{137}$Cs peak related to the Chernobyl nuclear accident in 1986, and (5) the sedimentary chemical signature (Mn, Fe, Hg) related to two major inflow of oxygenated North Sea water into the central Baltic Sea (Major Baltic Sea Inflows; Mohrholz et al., 2015) in 1994 and 2003.

We assigned an age of 2016 for the core top, i.e. the year of the core recovery, and an extrapolated age of 1860 to the core bottom. Estimated error bars for an age model based on an event stratigraphy are ca. 5-10 years and 1-2 years for the time markers, respectively, before and after 1960 (Kaiser et al., 2018). Linear sedimentation rates were assumed between the time markers. This dating method is the most adapted in marine environments, where the sediment composition can vary between homogenous (low organic carbon content) and laminated (high organic carbon content) over relatively short depth intervals

as it is the case for the Baltic Sea and the Black Sea (Moros et al., 2017; Kaiser et al., 2018). The mean sedimentation rate of core MSM51-2/20 increases constantly from ca. 0.2 cm/year before 1900s to ca. 0.7 cm/year towards the present-day in agreement with published data (Moros et al., 2017). In the upper 1-2 cm of the core, the sedimentation rate drops to 0.2 cm/year. This is an artefact related to the evaporation of the water from the fluff layer (> 95% of water) resulting in sediment compaction before core sampling. The mean sedimentation rate of core MSM51-2/20 is thus ca. 0.4 cm/yr. This means that

each sample represents an average of ca. 2-3 years.

## 2.3 Lipid analysis

Trap (n = 18) and core (n = 118) sediments were extracted using accelerated solvent extraction (Dionex ASE 350) with DCM/MeOH (9:1, v:v). After extraction, squalane was added as internal standard. The apolar lipid fractions were obtained by column chromatography ($SiO_2$) using hexane as eluent. They were analysed by gas chromatography-mass spectrometry

(GC-MS) using an Agilent Technologies 7890B GC system coupled to a 5977B Mass Selective Detector equipped with a HP-5ms capillary column (30 m x 0.25 mm x 0.25µm). The oven temperature was programmed from 40 to 320 °C at 8 °C/min followed by a 15 min isotherm. GC-MS data were collected in total ion current (TIC) (m/z 50–600). The apolar fractions were also analysed using a multichannel TraceUltra gas chromatograph (Thermo Fischer Scientific) equipped with a DB-5ms capillary column (30 m x 0.25 mm x 0.25µm) and a flame ionization detector (FID). The temperature program

was identical to that used for analysis by GC-MS and peak identification was made by comparison of retention indexes using the two methods. For quantification, the GC-FID response of each lipid was normalised to that of the internal standard and the amount of sediment extracted. The detection limit was estimated to ca. 5 ng/g. Because 6Me-$C_{17:0}$ and 7Me-$C_{17:0}$ monomethyl alkanes elute very close to each other on a DB-5ms capillary column, their total sum (6+7Me-$C_{17:0}$) was considered here. In order to correct for potential effects of lipid preservation/degradation and/or dilution by terrestrial inputs,

the data were normalized to total organic carbon (TOC) in both cores MSM51-2/20 and POS435/10 (TOC data published in Häusler et al., 2017) and the results are expressed as µg/gTOC. The data are listed in Tables S3, S4 and S5.





## 2.4 Monitoring data

The phytoplankton data were gathered according to the guidelines for the Baltic Monitoring Program of HELCOM
(HELCOM, 1988), which has been only slightly modified during the 38 years of the running monitoring programme (for
regular update see HELCOM, 2018). For representative sampling of the euphotic zone, the upper 10 m were considered, in
most cases by pooling discrete samples from 1, 2.5, 5, 7.5 and 10 m depth to one integrated sample. Acetic Lugol solution
was used as a fixative. The principles of the quantitative and qualitative microscopic analysis of phytoplankton are described
by Utermöhl (1958) and Olenina et al. (2006). The species list created by the HELCOM Phytoplankton Expert Group forms
the basis of unified species identification and biomass calculation in the Baltic Monitoring Program. The HELCOM
monitoring data including the IOW data are kept in the ICES data base and are freely available (www.ices.dk). They were
downloaded and restructured for our purpose. A total amount of 530 observations from 14 stations located in the EGB (Fig.
1; Table S1) were used here. The data for the years 2010–2011 are listed in Table S3, and the annual mean and summer
(July-August) biomass data for the period 1983–2016 in Table S4.

## 170    2.5 Frequency of Cyanobacteria Accumulations (FCA)

Quantitative estimates of the presence and extent of cyanobacteria blooms by satellite sensors are affected by the frequent
cloud cover. In order to normalize the detections of cyanobacteria accumulations to the amount of available cloud-free
imagery Kahru and co-workers (Kahru et al. 2007; Kahru and Elmgren 2014) introduced the Frequency of Cyanobacteria
Accumulations (FCA) that normalizes the number of cyanobacteria detections to the number of unobstructed views of a
satellite pixel. In 2010 the satellite observations were pooled from the MODIS-Aqua and MODIS-Terra sensors. FCA can be
calculated over variable periods. For the characterization of the annual bloom we used a period of two months (July -
August) and for the characterization of the seasonal dynamics we used 5 day intervals. FCA values for each 1 km$^2$ pixel were
averaged for the whole EGB region (Fig. 1; Table S3). The FCA data for the years 2010–2011 are listed in Table S3, and for
the period 1987–2016 in Table S4. These latter were published recently in Kahru et al. (2018).

## 180    2.6 Statistics

For the monitoring time interval (1983–2016), the biomass, FCA and biomarker data (Table S4) were processed without pre-
treatment. To be compared with the ca. 2-years resolved 6+7Me-C$_{17:0}$ record, the annually resolved HadISST1, AMO, and
NAO data (Table S5) were smoothed with a 2-point adjacent averaging, re-sampled with a 2-years step, normalized (min-
max normalization), and linearly detrended before computing the correlation coefficient using the PAST v3.25 software
(Hammer et al., 2001). Considering the different datasets and uncertainties related to the age model of core MSM51-2/20,
the correlation analyses were restricted to the period 1870–2006.





## 3 Results

In the EGB trap sediment time series from May 2010 to January 2011, 6+7Me-C$_{17:0}$ were found in May, July, August, early September, and early October 2010 (Fig. 3A). 6+7Me-C$_{17:0}$ fluxes ranged between ca. 5 and 200 µg/m$^2$/day with maxima in

July and August. Among the specific producers of 6Me-C$_{17:0}$ and 7Me-C$_{17:0}$ in the Baltic Sea (Bauersachs et al., 2017), the qualitative analysis of the main cyanobacterial genera in the trap sediments indicates the presence of *Aphanizomenon* sp. in May, July, August, late September, October, December 2010, and early January 2011, with abundance maxima in July and early October (Fig. 3B). *Nodularia spumigena* was observed in the trap sediments in July, August, early September, and early October 2010, with abundance maxima in July and August. Monitoring data based on stations from the EGB (Fig. 1)

display a cyanobacterial summer bloom in July-August with a highest total biomass in July (137 mg/m$^3$) (Fig. 3C). While *Aphanizomenon* sp. was present from May to November, *Nodularia spumigena* was present only in July-August. FCA data indicate that the bulk of cyanobacterial accumulations occurred over the whole month of July (Fig. 3D). The maximum FCA was detected during the 5-day period between July 10-14, when FCA reached nearly 40%, i.e. during this period nearly 40% of the satellite views of the sea surface in the EGB were classified as having accumulations.

Annual and summer (July-August) monitoring data of cyanobacterial biomass in the study area are highly positively correlated for the period 1983-2016 ($r^2$ = 0.97; n = 33; Table S4). For comparison purposes, the summer data are preferentially used here. The mean and standard deviation of the sum of *Nodularia spumigena* and *Aphanizomenon* sp. are 356 ± 632 mg/m$^3$ (Fig. 4A). Maxima occurred in the mid-1980s, in the early 1990s and in the mid-2000s. The highest total biomass value was recorded in 1986 (3139 mg/m$^3$). Biomass minima occurred in the early and the late 1980s, the mid-1990s,

and in the mid-2000s. The lowest value (20 mg/m$^3$) was registered in 2006.

In core MSM51-2/20, the mean and standard deviation values of 6+7Me-C$_{17:0}$ content are 0.5 ± 0.5 µg/gTOC (Figs. 4A-B and 5A). The record of 6+7Me-C$_{17:0}$ content displays a first increase in the early 1920s. Maximum contents occurred in the late 1930s, the mid-1950s, the early and late 1960s, the early 1970s, the mid- to late 1980s, the late 1990s, and in 2012, with values up to 2 µg/gTOC. The main minima were found in the late 1940s, the late 1950s, the mid-1960s, the late 1970s and

the late 1980s, the late 1990s, and the 2010s. The lowest values (0.02-0.05 µg/gTOC) occurred in the mid-1990s. In core POS435/10, the mean and standard deviation values of 6+7Me-C$_{17:0}$ content are 17 ± 60 µg/gTOC (Fig. 6A). Three main maxima are centred around 2,500, 4,500 and 6,500 years cal. BP and two main minima around 3,500 and 5,500 years cal. BP.

## 4 Discussion

**4.1 Diazotrophic cyanobacterial blooms in the central Baltic Sea in 2010-2011**

Highest fluxes of 6+7Me-C$_{17:0}$ in July-August 2010 reflect the main cyanobacterial blooming period characteristic for the central Baltic Sea as observed in trap sediments (Fig. 3). The occurrence of 6+7Me-C$_{17:0}$ fluxes correspond to an abundant to



highly abundant presence of both *Aphanizomenon* sp. and *Nodularia spumigena*. While *Aphanizomenon* sp. produces only relatively small amounts of 7Me-$C_{17:0}$, both 6Me-$C_{17:0}$ and 7Me-$C_{17:0}$ are produced in relatively high amount by *Nodularia*

*spumigena* (Bauersachs et al., 2017). However, 6Me-$C_{17:0}$ and 7Me-$C_{17:0}$ were present in May 2010 although *Nodularia spumigena* was not observed. *Aphanizomenon* sp. is very likely the source of 7Me-$C_{17:0}$, but the presence of 6Me-$C_{17:0}$ is somewhat puzzling. The absence of 6+7Me-$C_{17:0}$ in December 2010 and January 2011 although *Aphanizomenon* sp. was observed results very likely from both its low relative amount (mainly only "present") and its relatively low production of monomethyl alkanes (Bauersachs et al., 2017). The fluxes of 6+7Me-$C_{17:0}$ are not correlated to the relative amounts of

*Aphanizomenon* sp. (r = 0.19; p = 0.66; n = 8), but are slightly positively correlated to *Nodularia spumigena* (r = 0.53; p = 0.28; n = 6). Both the biomass data and FCA data suggest that the main phase of the cyanobacterial bloom occurred in July, with a maximum extent of ca. 70,000 km$^2$ estimated on the 12$^{th}$ of July (Hansson and Öberg, 2010), and decreased in August. Relatively high abundance of cyanobacteria in trap sediments as late as early September and early October maybe related to a delayed signal between the water surface and the sediment trap related to lower sinking velocities after the main

phase of the bloom. Despite this possible delay, the 6+7Me-$C_{17:0}$ content of the sediment is likely reflecting semi-quantitatively the summer bloom of *Nodularia spumigena*, which is usually the most abundant diazotrophic cyanobacterial strain in central Baltic Sea summer blooms (Kononen, 1992; Finni et al., 2001; Wasmund et al., 2018).

**4.2 Diazotrophic cyanobacterial blooms in the central Baltic Sea between 1983-2016**

In order to further constrain 6+7Me-$C_{17:0}$ as potential cyanobacterial biomarker for the Baltic Sea, the 6+7Me-$C_{17:0}$ contents

of the dated core MSM51-2/20 were compared to both July-August monitoring and satellite-derived FCA data (Fig. 4). Monitoring and satellite (FCA) data are not easy to compare on a long-term trend for two main reasons. First, monitoring data are punctual in space and time, while the satellite-based data used here are integrated on a basin-wide scale. Second, the satellite data used here reflect mostly the surface and near-surface accumulations and not so much of the organisms thriving deeper in the water column (Kahru and Elmgren, 2014), while monitoring data integrate the upper 10 m of the water column.

However, when removing the exceptionally high biomass of *Nodularia spumigena* in 1992 (2149 mg/m$^3$; Table S4), the FCA index is significantly correlated to the biomass of *Nodularia spumigena* (r = 0.61; p < 0.01; n = 27), but not to the biomass of *Aphanizomenon* sp. (r = 0.01; p = 0.97; n = 27). This can be expected as *Nodularia spumigena* concentrates at the water surface (Eigemann et al., 2018), whereas *Aphanizomenon* spp. may accumulate in subsurface water layers (Hajdu et al., 2007). The 6+7Me-$C_{17:0}$ content is significantly positively correlated neither to the FCA index (r = 0.08; p = 0.71; n =

22), nor to the biomass of *Nodularia spumigena* (r = 0.10; p = 0.62; n = 26), nor to the biomass of *Aphanizomenon* sp. (r = -0.36; p = 0.07; n = 26). This is very likely because each sediment sample integrates 2-3 years, and because the age uncertainties of the biomarker-based record are large (± 1-2 years) compared to satellite and monitoring observations. However, considering this age uncertainty, it seems that 6+7Me-$C_{17:0}$ contents are relatively high (low) when both *Nodularia spumigena* biomass and the FCA index are high (low), e.g. between the early 1990s and the mid-2000s. Therefore, we





suggest that 6+7Me-$C_{17:0}$ is likely reflecting semi-quantitatively fluctuations in cyanobacterial biomass and accumulations in the EGB as already indicated by the sediment trap data.

### 4.3 Diazotrophic cyanobacterial blooms in the central Baltic Sea over the last 140 years

Based on a compilation of published historical records, Finni et al. (2001) could provide a discontinuous record of cyanobacterial open-sea blooms before the beginning of Baltic Sea monitoring programs. In the early 20[th] century,
cyanobacterial accumulations were rare, what may be explained by relatively cool to moderate summers. Blooms occurred abundantly in successive summers between 1924 and 1930. Favourable weather conditions with several warm summers promoted strong blooms in the 1970s (Kononen and Niemi, 1984). These observations are in agreement with the 6+7Me-$C_{17:0}$ record, which suggests a quasi-absence of diazotrophic cyanobacteria (and likely mainly *Nodularia spumigena*) in the central Baltic Sea between 1860-1910, in the late 1940s, the late 1950s, and the mid-1960s, but a relatively high abundance
in the 1920s-1930s, in the mid-1950s, in the early and late 1960s, and in the early 1970s (Fig. 5A). While it has been suggested that the massive anthropogenic loading of nutrients to the Baltic Sea increased cyanobacterial productivity (Finni et al., 2001; Poutanen and Nikkilä, 2001; Stal et al., 2003; Vahtera et al., 2007), it is interesting to note here that no significant increase in cyanobacteria occurred in the 1950s, when nutrient inputs, eutrophication and deoxygenation of the Baltic Sea became explicit (Zillèn et al., 2008; Savchuk et al. 2012; Carstensen et al., 2014). Our data provide the first,
continuous record of cyanobacterial activity in the central Baltic Sea at a sub-decadal resolution beyond instrumental data, and allows analysing its decadal to multi-decadal variability over the last 140 years.

A positive correlation exists between fluctuations in cyanobacterial biomass in the central Baltic Sea as reconstructed with the 6+7Me-$C_{17:0}$ proxy and Baltic Sea summer (July-August) sea surface temperature at a decadal to multi-decadal timescale ($r = 0.25$; $p < 0.05$; $n = 70$; Table S6), i.e. higher (lower) cyanobacterial blooms during years with higher (lower) temperature
(Fig. 5A-B). Although temperature has been suggested as a key factor that triggers the formation of a bloom (Sellner, 1997), temperature does not affect the euphotic depth (below which cyanobacteria are unable to perform net photosynthesis) and the critical depth (below which the depth-integrated cyanobacterial primary productivity is zero) of the cyanobacterial community (Stal et al., 2003). Instead, changes in solar irradiance reaching the surface and in the angle of the sun affect the critical depth. Indeed, years with more sunshine in July-August tend to have more cyanobacterial accumulation (Kahru et al.,
1994). Temperature is, however, an important contributory factor since it causes a stabilization of the water column and decreases the mixing depth, thereby increasing the light irradiance available for the cyanobacteria (Stal et al., 2003). Recently, Kniebusch et al. (2019) and Börgel et al. (2018) found that the Atlantic Multidecadal Oscillation (AMO; Fig. C; Knight et al., 2006) causes changes in Baltic Sea surface temperature. The AMO, which is defined as the alternation of cold and warm phases in North Atlantic surface temperature with a period of 60-90 years, is positively correlated to the 6+7Me-
$C_{17:0}$ ($r = 0.30$; $p < 0.05$; $n = 70$; Table S6). Therefore, the AMO is likely a factor determining the variability of cyanobacterial biomass in the central Baltic Sea at a multi-decadal timescale. The North Atlantic Oscillation (NAO; Hurrell et al., 2003) is another large-scale climate mode and it dominates winter climate variability in Europe. At a multi-annual to





decadal timescale, the NAO is determining to a large extent ice cover, surface wind, freshwater runoff, and temperature in the Baltic region (Hänninen et al., 2000; Omstedt and Chen, 2001; Kauker and Meier, 2003). Using computer simulations, Janssen et al. (2004) proposed a cause-and-effect chain between the NAO and cyanobacterial blooms. Positive NAO leads to low ice cover and high wind stress triggering a deepening of the surface mixed-layer and an increase in excess dissolved inorganic phosphorus, favouring late summer cyanobacterial blooms during the following summer. However, the 6+7Me-$C_{17:0}$ record is not correlated to the NAO index over the last 140 years (r = 0.10; p = 0.43; n = 70; Table S6). Therefore, our data suggest that fluctuations in the cyanobacterial biomass in the central Baltic Sea at a decadal to multi-decadal timescale are at least partly related to sea surface temperature changes, which are ultimately triggered by the AMO.

## 4.4 Diazotrophic cyanobacterial blooms in the Bothnian Sea during the Holocene

6+7Me-$C_{17:0}$ contents in the Bothnian Sea during the Holocene Thermal Maximum (HTM; 4,500–8,000 years cal. BP *sensu lato*; Borzenkova et al., 2015) were up to 100-fold higher than in the modern central Baltic Sea (Figs. 4 and 5A). This suggests a high abundance of diazotrophic cyanobacteria (at least regarding *Nodularia spumigena*) in a region of the Baltic Sea, where the diazotrophic cyanobacterial biomass is ca. 4- to 5-fold lower than in the central Baltic Sea in the present day (Wasmund et al., 2018). An abundant presence of cyanobacteria in the central and northern Baltic Sea during the HTM has already been shown in previous studies (Bianchi et al., 2000; Funkey et al., 2014; Jilbert et al., 2015; Sollai et al., 2017). This pattern is very likely related to three main factors. First, the volume of the Baltic Sea was much larger than today due to glacial overdeepening and early Holocene eustatic sea-level rise resulting in a stronger influence of North Sea water and, very likely, a pronounced water column stratification (Jilbert et al., 2015; Häusler et al., 2017). Second, temperature and insolation were higher-than-today during the HTM (Seppä et al., 2005, 2009; Borzenkova et al., 2015). Third, intensely hypoxic to anoxic conditions during the HTM may have stimulated the sediment-bound phosphorus to be released into the water column (Vahtera et al., 2007; Jilbert et al., 2013; 2015; Funkey et al., 2014). These conditions very likely provided an ideal environment for diazotrophic cyanobacteria to thrive. In turn, frequent cyanobacterial blooms may have contributed to the formation of hypoxic to anoxic deep waters (Jilbert et al., 2015; Häusler et al., 2017), as it is the case for the central Baltic Sea in the present-day (Zillen and Conley, 2010). The 6+7Me-$C_{17:0}$ record suggests a fluctuating abundance of diazotrophic cyanobacteria between 0–3,000 years before present. This pattern is absent in a previous study using echinenone and zeaxanthin pigments to reconstruct cyanobacterial blooms in a sediment core from the Bothnian Sea (Jilbert et al., 2015). The non-specificity to diazotrophic cyanobacteria (Bianchi, 2007) and the diagenetic instability (Britton et al., 2008) of these organic biomarkers may explain the discrepancies between both records. The variability in diazotrophic cyanobacterial blooms in the Bothnian Sea over the last 7,000 years seems to be related to mean annual temperature changes as recorded in lake sediments from southern and central Sweden (Fig. 6B; Seppä et al., 2009). Periods with relatively higher temperature co-occur with a higher presence of cyanobacterial lipids in the sediments, that may be related to the enhancement of cyanobacteria blooms by higher temperature (Stal et al., 2003). The present results indicate that 6+7Me-$C_{17:0}$ is a relevant proxy for diazotrophic cyanobacterial blooms in the Baltic Sea for the whole duration of the Holocene.
# 5 Conclusions

Using in situ monitoring data, satellite data, sediment traps, and sediment cores from the EGB it was possible to show that the $6+7Me\text{-}C_{17:0}$ content of the sediment is very likely reflecting semi-quantitatively the biomass of diazotrophic

cyanobacteria, and more especially of *Nodularia spumigena*, which is the most abundant diazotrophic cyanobacterial species in the central Baltic Sea summer bloom. By applying this biomarker to a short sediment core, a continuous reconstruction of diazotrophic cyanobacterial abundance in the central Baltic Sea with a sub-decadal resolution and extending beyond instrumental data was provided for the first time. This record suggests that cyanobacterial blooms have not increased due to anthropogenic nutrient loading. Cyanobacterial biomass fluctuation seems to be at least partly related to sea surface

temperature changes and the AMO climate mode at a decadal to multi-decadal timescale over the last 140 years. The application of the $6+7Me\text{-}C_{17:0}$ biomarker on a long sediment core from the northern Baltic Sea revealed that the variability in cyanobacterial blooms was also related to temperature variability for the whole duration of the Holocene. The present results are of significance for the Baltic Sea ecosystem when considering the on-going global warming.

## Supplements

Tables S1 to S6.

## Author contribution

J.K., N.W., and M.K. wrote the manuscript; J.K., A.K.W., and K.H. produced the biomarker data, N.W. and R.H. the biomass data, and M.K. the satellite data; M.M. provided the $^{137}$Cs data; D.S.-B. provided the PCBs data; H.W.A. and all co-authors contributed to the discussion and commented on the manuscript.

## Competing interests


The authors declare that they have no conflict of interest.

## Acknowledgments

We are extremely grateful to Falk Pollehne for his comments on an earlier version of the manuscript. Nadine Hollmann is thanked for lab work. This work was supported by the German Research Foundation (grant KA 3228/2-1 to J.K.) and the

Leibniz Association (grant SAW-2017-IOW-2 to H.W.A).



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

**Figure captions**

**Figure 1.** Location of the monitoring stations (black dots), the sediment trap (triangle), the short sediment cores MSM51-2/20 and
EMB1215/7 (circle), and the long sediment core P435/10 (circle) investigated in the present study. The dotted lines represent the limits of the area considered to determine the frequency of cyanobacterial accumulations index of the Eastern Gotland Basin (Kahru et al., 2018). The locations of Lake Flarken and Lake Klotjärnen (Seppä et al., 2009) are also shown (stars). The Ocean Data View software (Schlitzer, 2016) was used to create the map and plot the data. See also Table S1.

**Figure 2.** Age models of short sediment cores EMB1215/7 (grey curves) and MSM51-2/20 (black curves) based on records of pollutants
(mercury, Hg; sum of polychlorinated biphenyls, PCBs), radionuclides (unsupported $^{210}$Lead, $^{210}$Pb$_{unsupp.}$; $^{241}$Americium, $^{241}$Am; $^{137}$Caesium, $^{137}$Cs), and XRF-scanning elements (Mn in counts per second, and the inc/coh ratio). The dashed lines represent core EMB1215/7 depths dated by event stratigraphy (year AD) as published in Moros et al. (2017) (except for the PCBs record) and their equivalent depths in core MSM51-2/20. The data are listed in Table S2.

**Figure 3.** May 2010 to January 2011 fluxes of the sum of 6Me-C$_{17:0}$ and 7Me-C$_{17:0}$ (6+7Me-C$_{17:0}$) (A), relative abundance (1: present; 2:
abundant; 3: highly abundant; stacked bars) of *Nodularia spumigena* and *Aphanizomenon* sp. cyanobacteria in trap sediments from the Eastern Gotland Basin (B), biomass data of *Nodularia spumigena* and *Aphanizomenon* sp. cyanobacteria from monitoring stations located in the Eastern Gotland Basin (Fig. 1) (C), and the FCA (Frequency of Cyanobacteria Accumulations) index (D). The data are listed in Table S3.

**Figure 4.** Core MSM51-2/20 record of the sum of 6Me-C$_{17:0}$ and 7Me-C$_{17:0}$ (6+7Me-C$_{17:0}$) normalized to TOC (line with dots) (A, B), and
bar plots of *Nodularia spumigena* and *Aphanizomenon* sp. summer (July-August) biomass (A) and of the FCA (Frequency of Cyanobacteria Accumulations) index (Kahru et al., 2018) (B), both for the Eastern Gotland Basin (Fig. 1). The diamonds represent the age control points of core MSM51-2/20 (Fig. 2; Table S2). The data are listed in Table S4.

**Figure 5.** Core MSM51-2/20 record of the sum of 6Me-C$_{17:0}$ and 7Me-C$_{17:0}$ (6+7Me-C$_{17:0}$) normalized to TOC (line with dots) (A), summer (July-August) HadISST1 (Rayner et al., 2003) sea surface temperature (SST) for the Baltic Sea (B), the Atlantic Multidecadal
Oscillation index (AMO; Enfield et al., 2001; https://www.esrl.noaa.gov/psd/data/timeseries/AMO/) (C), the winter (DJFM) North Atlantic Oscillation index (NAO; Hurrell, 1995; Jones et al., 1997; https://www.esrl.noaa.gov/psd/gcos_wgsp/Timeseries/NAO/) (D). Both raw (light grey) and 5-pts adjacent averaged (black line) data are shown. The diamonds represent the age control points of core MSM51-2/20 (Fig. 2; Table S2). The horizontal grey rectangles represent two periods with relatively high cyanobacterial blooms as inferred from historical records (Finni et al., 2001). The vertical dashed lines highlight the periods with low 6+7Me-C$_{17:0}$ content. The data are listed in
Table S5.

**Figure 6.** Holocene (0–7,000 years cal. BP) records of the TOC normalized sum of 6Me-C$_{17:0}$ and 7Me-C$_{17:0}$ (6+7Me-C$_{17:0}$; log scale) in the Bothnian Sea (core POS435/10; Häusler et al., 2017) (A) and of two pollen-based annual mean temperature reconstructions from Swedish lakes (Seppä et al., 2009) (B). The Littorina transgression marking the limit between the Ancylus Lake stage and the Littorina Sea stage is shown as dashed line. The data are listed in Table S7.







**Figure 1**

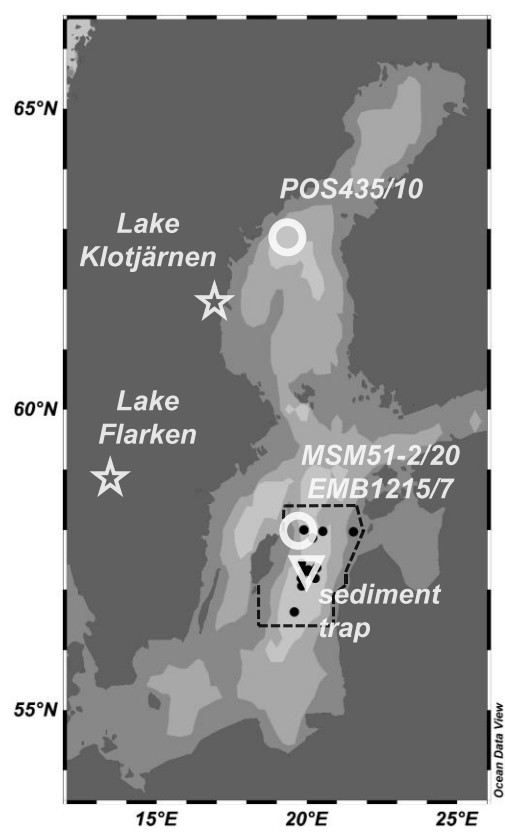






**Figure 2**


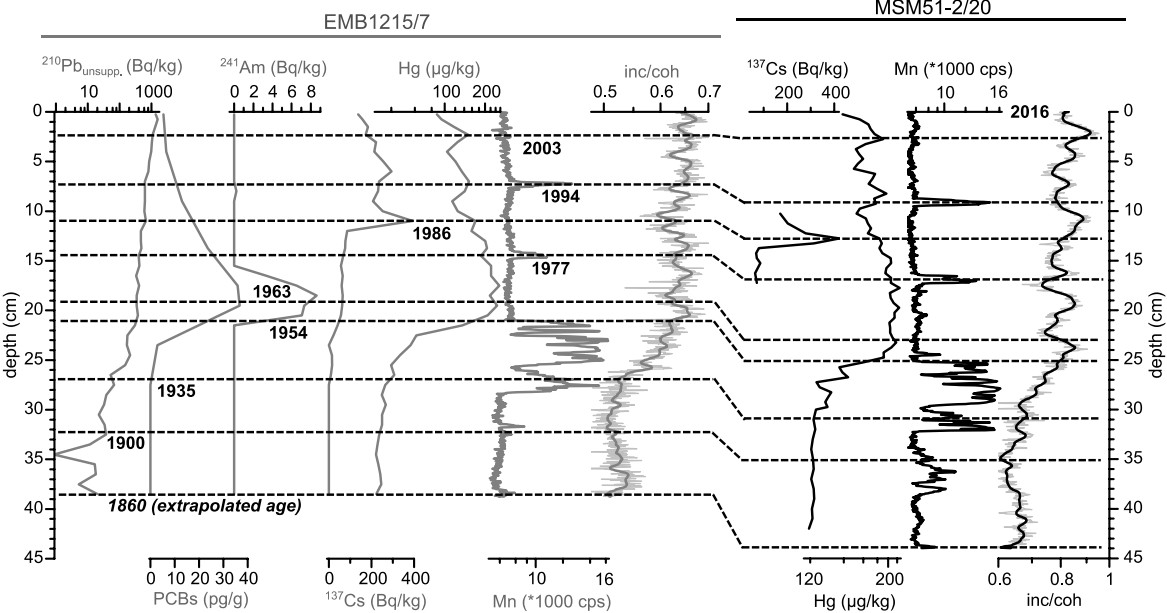






**Figure 3**

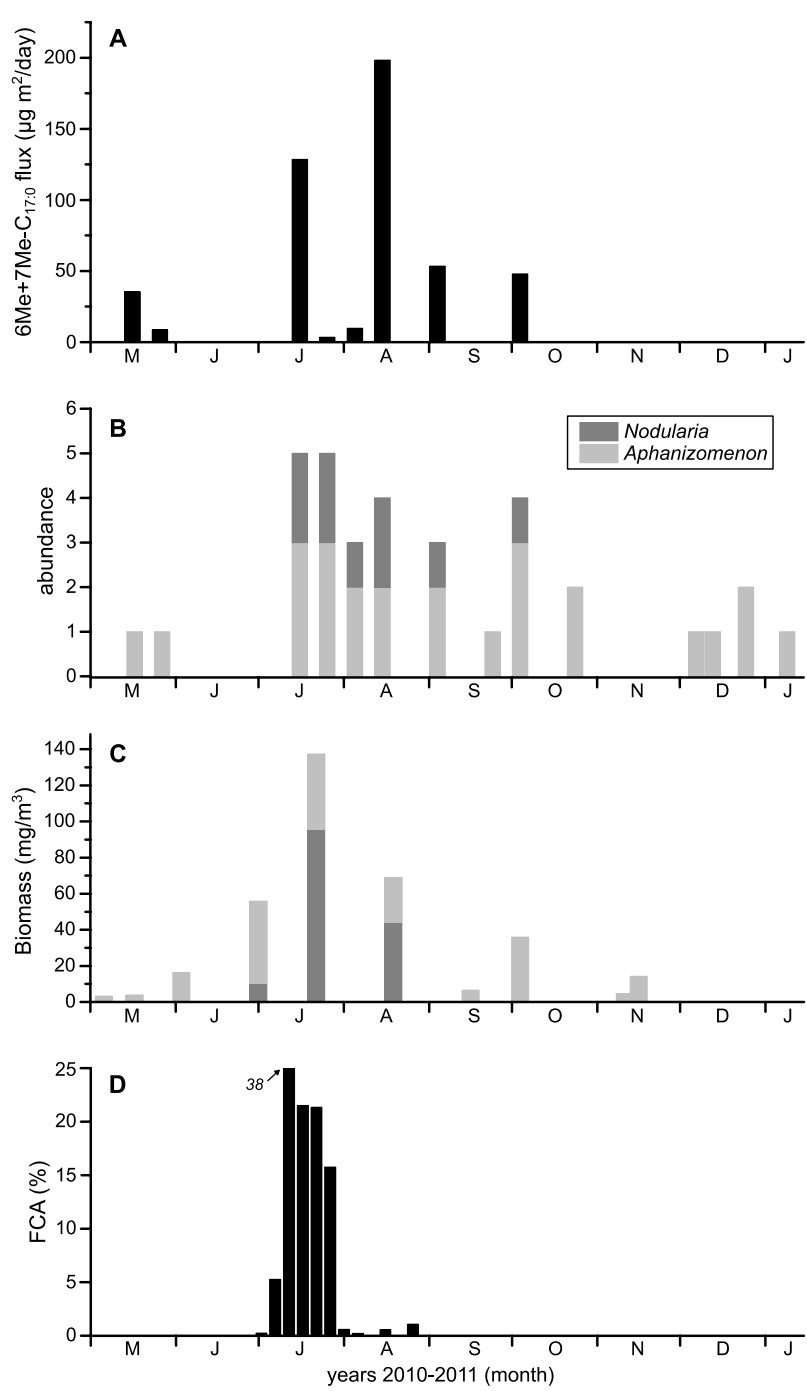



**Figure 4**

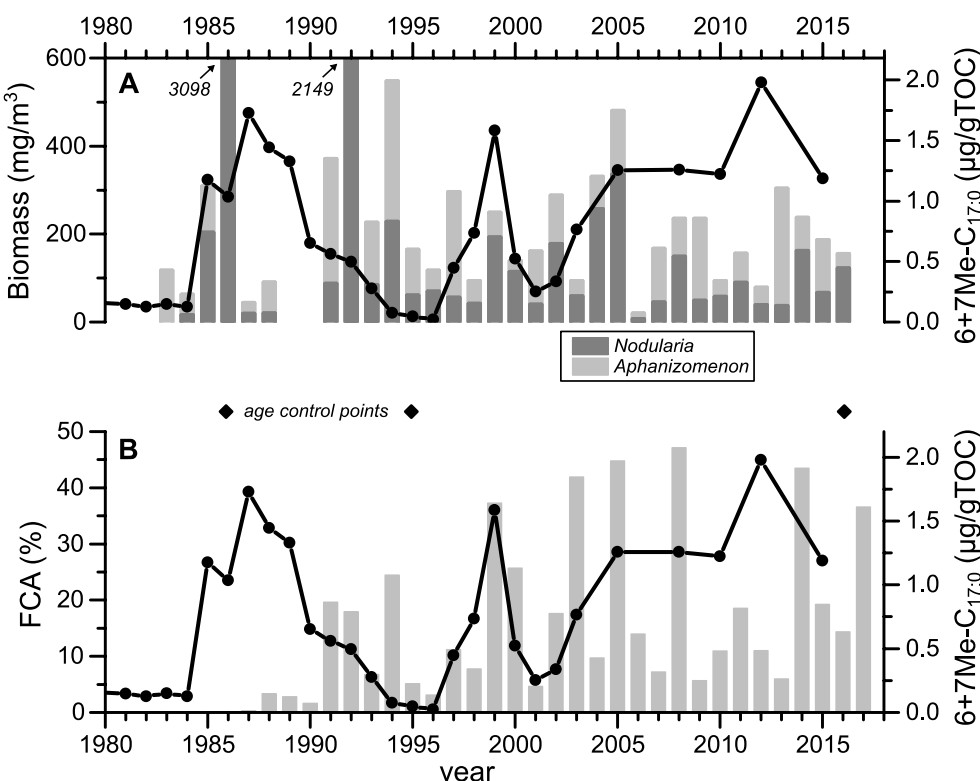








**Figure 5**


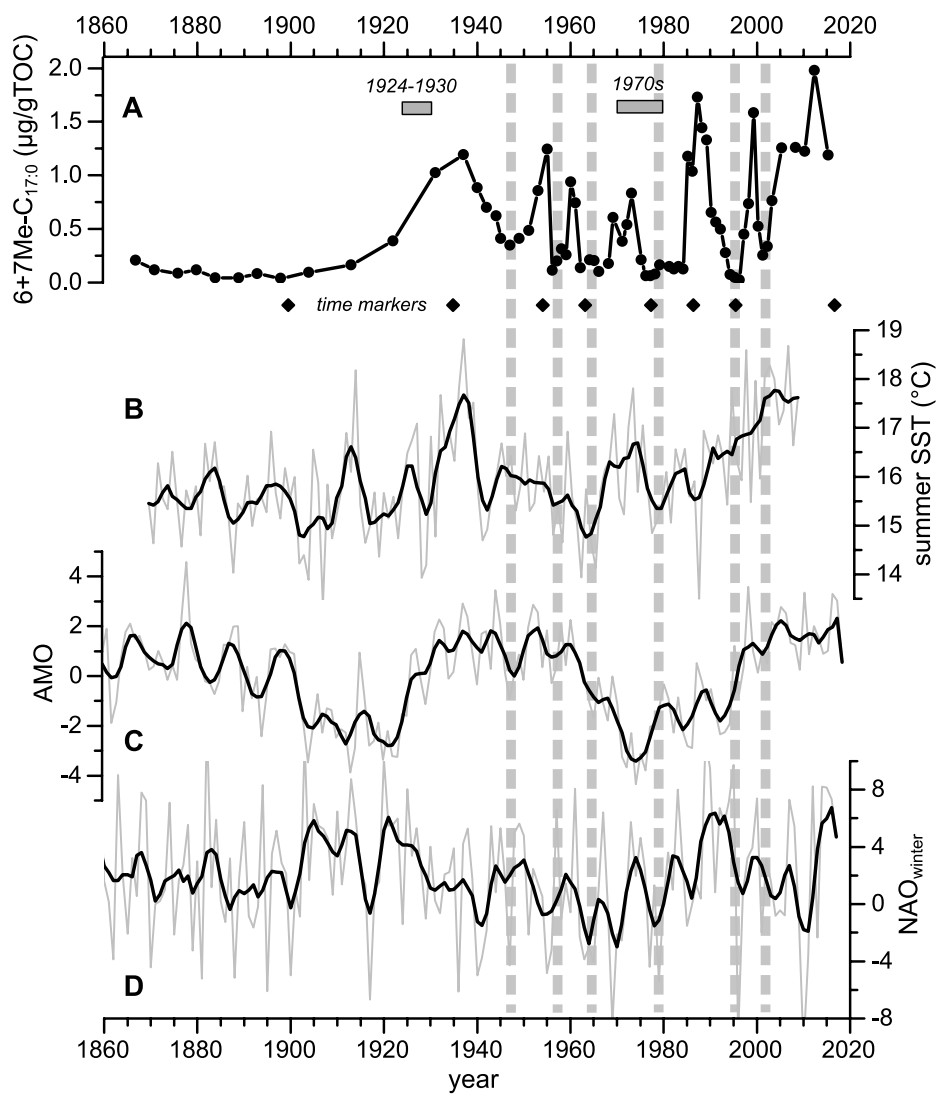


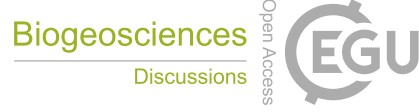

**Figure 6**


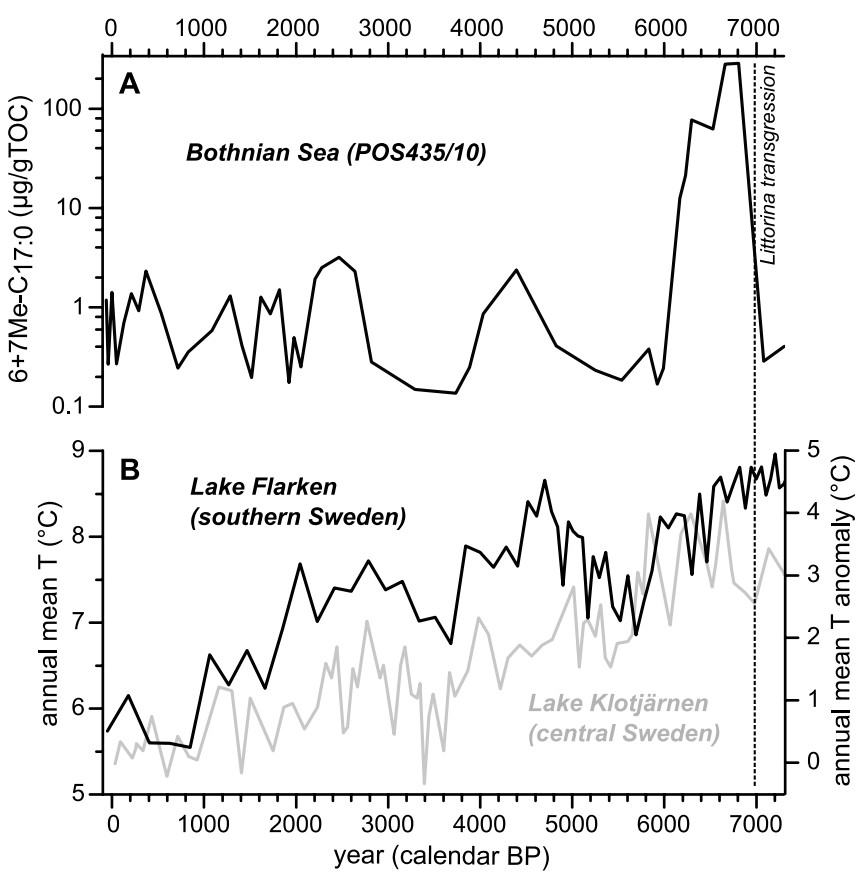