# Peer review of "Reconstructing N2-fixing cyanobacterial blooms in the Baltic Sea beyond observations using 6- and 7-methylheptadecanes in sediments as specific biomarkers"

_Biogeosciences, 2019_

## Referee Comment (RC1) · Anonymous Referee #1 · 17 Dec 2019

The paper by Kaiser et al. is well written. The authors take a systematic approach to applying 6 and 7-methylheptadecane (6+7Me-C17:0) as a biomarker for Nodularia cyanobacteria. They first analyzed 6+7Me-C17:0 in sediment traps from the Baltic and then in a series of sediment cores from nearby locations. In a core from 1860 – the present day the concentration of 6+7Me-C17:0 correlated well with the Baltic Sea SST at a decadal to multi-decadal timescale and with the AMO.

There are some issues that I believe need to be addressed before this paper can be accepted for publication. Unfortunately I do not agree that the authors have sufficiently

shown that the sum of 6 and 7- Me-C17:0 is a "robust semi-quantitative biomarker for cyanobacteria" in the sediment trap samples or in the 1980 – 2015 section of the MSM51-2/20 core. Indeed, there are several mismatches in the sediment trap data and the MSM51-2/20 core data between 6+7Me-C17:0 and the presence of Nodularia. This raised concerns for me as the authors selected the sum of 6- and 7-Me-C17:0 as they were both detected in 4 strains of Nodularia by Bauersachs et al. (2017).

I think it is important to note that Bauersachs et al. report a range of 10 hydrocarbons across 8 species of Baltic Sea cyanobacteria, from the genera Dolichospermum, Aphanizomenon and Nodularia. I feel that it would be more informative to present all the hydrocarbon data from the sediment traps samples, not just 6+7Me-C17:0. Information about the presence/absence of n-C17 and other monomethyl alkanes (MMAs) or dimethyl alkanes (DMAs) would be helpful (and really interesting). This full distribution could be compared (statistically) with a wider range of cell counts of e.g. Dolichospermum, Aphanizomenon and Nodularia species. This would provide a solid base for going forward with 6+7Me-C17:0 as a marker for Nodularia, if the data supports it. For example, I notice that Bauersachs et al. reported that Nodularia was the only genera that also produced DMAs. If those components were also found along with the 6+7Me-C17:0 it would make a much stronger argument for applying 6+7Me-C17:0 as a biomarker for the occurrence of Nodularia in the past Baltic Sea.

Furthermore, it is unfortunate that the separate abundances of 6- and 7- Me-C17:0 were not reported. In the Bauersachs et al. paper the 6 Me-C17:0 to 7- Me-C17:0 ratio was consistently around 0.25 in all 4 Nodularia species. For all the reader knows, the sum in this study could consist primarily of 6-methylheptadecane, which would not correspond to any of the profiles found by Bauersachs et al. Using GC-MS, it should be possible to estimate the proportion of 6 Me-C17:0 and 7- Me-C17:0 by integrating both components from their mass chromatograms. If the 6- and 7-methylheptadecanes come from a different source then a different ratio could be expected.

As I said before, I found the paper well written and interesting but, without a more

robust connection between the biomarker and its cyanobacterial source, the extension of the study to the sediment cores carries a high risk. There is too much uncertainty arising from the fact the 6+7Me-C17:0 concentrations only show very low correlation with FCA and the Nodularia and Aphanizomenon biomass data.

Additional points to address:

Abstract Line 10 – Change "time history" to "history"

Line 12 – Change "trap sediments" to " sediment traps"

Line 19 – Remove "rather"

Introduction

Lines 32 – 34 (and at other points in the manuscript). It does not read well to have a list of items, separated by commas but with multiple cases of "and" as the conjunction. For example, I would write the sentence as: "If diazotrophic cyanobacteria occur in large blooms they contribute to nitrogen-eutrophication, where the massive export and decay of cyanobacterial biomass results in O2 consumption, leading to the spread of bottom water hypoxia and anoxia (Zillen and Conley, 2010; Feistel et al., 2016)."

Line 61 – Change to "considered well suited"

Discussion Line 244 – 246 – Strange wording. I would change to "The 6+7Me-C17:0 content is not significantly positively correlated to the FCA index (r = 0.08; p = 0.71; n =245 22), nor to the biomass of Nodularia spumigena (r = 0.10; p = 0.62; n = 26), nor to the biomass of Aphanizomenon sp. (r = -0.36; p = 0.07; n = 26)."

Line 255 – Should be "which may be"

Line 277 – Figure number missing.

---

## Referee Comment (RC2) · Tom Jilbert (Referee) · 23 Dec 2019

**Interactive comment by Tom Jilbert on "Reconstructing N2-fixing cyanobacterial blooms in the Baltic Sea beyond observations using 6- and 7-methylheptadecanes in sediments as specific biomarkers" by Kaiser et al.**

This manuscript presents several interesting datasets concerning the past abundance of diazotrophic cyanobacteria in the Baltic and Bothnian Seas on various timescales. The datasets are derived both from direct observations (water column- and sediment trap- monitoring of genera abundance, as well as satellite-based observations of bloom frequency) and from a new organic proxy in sediment trap and core samples, namely the abundance of mid-chain branched alkane (6+7Me-C17:0) lipids. The main goal of the study is test the applicability of these biomarkers for the reconstruction of past diazotrophic cyanobacterial abundance, and indeed the authors present one such long sediment core record from the Bothnian Sea. The authors also use their biomarker data, along with instrumental and proxy-derived time series of climatic parameters, to investigate the potential climatic forcing of bloom occurrence on various timescales. The intrinsic value of the proxy seems to be high, and the paper is well written. However I have concerns over the authors' conclusions about the drivers of cyanobacterial bloom occurrence on various timescales, in particular their strong favoring of temperature over nutrient dynamics.

**Major comments**

1. The conclusion stated several times in the manuscript, e.g. Line 323, "This record suggests that cyanobacterial blooms have not increased due to anthropogenic nutrient loading" is too bold considering the data presented in the study. Most researchers would agree that cyanobacterial bloom occurrence during recent decades is influenced by both temperature and nutrient dynamics, so without a strong piece of evidence to refute one or other factor, I suggest to moderate the wording of these sections. I also suggest that the authors should add the 20[th] century nutrient loading time series to Figure 5, in order for the reader to see how this compares with the other data presented. Some further considerations related to this:

- The principal reasoning for stating that blooms respond more to temperature than nutrient loading is the "early onset" of blooms in the 20[th] century as implied by the peak in 6+7Me-C17:0 lipids in the period 1920-1940 in the Gotland Basin core. However monitoring data show that phosphorus loading during this period increased by some 20% with respect to the period 1900-1920 (see below)

[Figure]

*HELCOM data of phosphorus loading to the sub-basins of the Baltic Sea, 1900-2010. Mean values for 1900-1920 and 1920-1940 are shown in red.*

This increase of 20% is greater in absolute terms (and certainly less noisy) than the SST increase over the same period (approx. 1°C +/- 2°C, see below)

[Figure]

*Biomarker and SST time series presented in the study. Mean SST values for 1900-1920 and 1920-1940 periods are indicated by red bars.*

Of course, in the case of both nutrient loading and temperature, the response of cyanobacterial blooms is expected to be non-linear, e.g. a threshold-type response, related to the fact that these organisms are competing for resources within an ecosystem, and above certain thresholds of certain environmental variables may gain significant competitive advantage over other primary producers. Hence the difficulty in making direct linear correlation analyses with time series of those environmental variables. In summary, I would like to see a more balanced acknowledgement that both nutrient loading and temperature may have influenced bloom occurrence during this period, and that these responses are likely strongly non-linear.

2. I have a similar concern with the interpretation of the long sediment core record in Figure 6, although now we are discussing natural rather than anthropogenically-impacted nutrient cycling. The authors acknowledge in the text that phosphorus regeneration played an important role in sustaining blooms during the HTM in the Bothnian Sea, as we showed in our earlier study (Jilbert et al., 2015). However I would also like to see a statement acknowledging that declining P availability was likely the main factor in the steep decline in blooms from 6500 yr. B.P., which is a dominant feature of this record (we interpreted this as due to the shoaling of the Åland Sea sills). The temperature records from the Swedish lakes presented here support the concept of warm conditions favoring blooms during the HTM, but for example, 4500 yr. B.P. shows a similar temperature to 6500 yr. B.P., yet the bloom intensity is orders of magnitude lower as shown by the log-scale of 6+7Me-C17:0. This requires another controlling factor, ie. availability of P.

**Minor comments**

Line 96: Replace 'bloom' with 'blooms'

Line 97: One could reasonably ask why a core from the Bothnian Sea is investigated and not from the same location as the short cores and sediment trap series

Line 106: Give more detail on the coring device

Line 111: Methodology for estimating TOC needs more detail. Is one of these instruments able to isolate and measure inorganic carbon from a bulk sample?

Line 255: Replace 'what' with 'which'

---

## Author Comment (AC1) · 7 Feb 2020

Anonymous Referee #1 The paper by Kaiser et al. is well written. The authors take a systematic approach to applying 6 and 7-methylheptadecane (6+7Me-C17:0) as a biomarker for Nodularia cyanobacteria. They first analyzed 6+7Me-C17:0 in sediment traps from the Baltic and then in a series of sediment cores from nearby locations. In a core from 1860 – the present day the concentration of 6+7Me-C17:0 correlated well with the Baltic Sea SST at a decadal to multi-decadal timescale and with the AMO. There are some issues that I believe need to be addressed before this paper can be

accepted for publication. Unfortunately I do not agree that the authors have sufficiently shown that the sum of 6 and 7- Me-C17:0 is a "robust semi-quantitative biomarker for cyanobacteria" in the sediment trap samples or in the 1980 – 2015 section of the MSM51-2/20 core. Indeed, there are several mismatches in the sediment trap data and the MSM51-2/20 core data between 6+7Me-C17:0 and the presence of Nodularia.

Answer: We agree that there are several mismatches between the lipids in the trap and short core sediments and the presence of Nodularia. Indeed, there is a large amount of potential error sources as mentioned in the manuscript. But, based on the trap sediment data, and considering the age model error bars of the short sediment core, we still think that the sum of 6Me- and 7Me-C17:0 can be considered as potential semi-quantitative biomarkers for cyanobacteria; not for all cyanobacteria, but specifically for Nodularia. However, we have moderated the wording: we do not mention "robust" anymore in the text, and we have added "potentially" instead of "likely" and "very likely".

This raised concerns for me as the authors selected the sum of 6- and 7-Me-C17:0 as they were both detected in 4 strains of Nodularia by Bauersachs et al. (2017). I think it is important to note that Bauersachs et al. report a range of 10 hydrocarbons across 8 species of Baltic Sea cyanobacteria, from the genera Dolichospermum, Aphanizomenon and Nodularia. I feel that it would be more informative to present all the hydrocarbon data from the sediment traps samples, not just 6+7Me-C17:0. Information about the presence/absence of n-C17 and other monomethyl alkanes (MMAs) or dimethyl alkanes (DMAs) would be helpful (and really interesting). This full distribution could be compared (statistically) with a wider range of cell counts of e.g. Dolichospermum, Aphanizomenon and Nodularia species. This would provide a solid base for going forward with 6+7Me-C17:0 as a marker for Nodularia, if the data supports it. For example, I notice that Bauersachs et al. reported that Nodularia was the only genera that also produced DMAs. If those components were also found along with the 6+7Me-C17:0 it would make a much stronger argument for applying 6+7Me-C17:0 as a biomarker for the occurrence of Nodularia in the past Baltic Sea.

[Figure]

Answer: We have looked again carefully at the GC-MS data in case of possible error, but we came to the same results that the DMAs produced by Nodularia were not present in the samples, likely because of both a relatively low production and lipid degradation. Concerning MMAs, we didn't consider the MMAs (n-C16:0, n-C17:0) as these are not specific biomarkers, and can have many other sources than cyanobacteria in sediments. Furthermore, there is no correspondence when comparing n-C17:0 fluxes in trap sediments and the relative amounts of Aphanizomenon and Dolichospermum (as main producers following Bauersachs et al., 2017). As we are focusing on potential specific biomarkers for cyanobacteria in the present study, we prefer not to include n-C17:0 data to avoid confusion. The ubiquitous source of n-C17:0 alkane has been mentioned in the Introduction. The data are attached for Reviewer #1 (Table for R#1), but will not be published here. Finally, cell counts of Dolichospermum, Aphanizomenon, and Nodularia are unfortunately not available for the trap sediments so that a statistical comparison with the lipid distribution is not possible. The following sentence has now been added in the Results: "Bauersachs et al. (2017) found five monomethyl alkanes (6Me-C17:0, 7Me-C17:0, 8Me-C17:0, 7Me-C16:0 and 7Me-C15:0) and two dimethyl alkanes (6,12Me-C17:0 and 7,11Me-C17:0) in cultures of cyanobacteria strains belonging to the genera Dolichospermum, Aphanizomenon and Nodularia isolated from the Baltic Sea. However, no monomethyl alkanes other than 6Me-C17:0 and 7Me-C17:0 nor dimethyl alkanes were found in the sediments. This may be due to a relatively low production of these lipids and/or a poor preservation."

Furthermore, it is unfortunate that the separate abundances of 6- and 7- Me-C17:0 were not reported. In the Bauersachs et al. paper the 6 Me-C17:0 to 7- Me-C17:0 ratio was consistently around 0.25 in all 4 Nodularia species. For all the reader knows, the sum in this study could consist primarily of 6-methylheptadecane, which would not correspond to any of the profiles found by Bauersachs et al. Using GC-MS, it should be possible to estimate the proportion of 6 Me-C17:0 and 7- Me-C17:0 by integrating both components from their mass chromatograms. If the 6- and 7-methylheptadecanes come from a different source then a different ratio could be expected. As I said before,

I found the paper well written and interesting but, without a more robust connection between the biomarker and its cyanobacterial source, the extension of the study to the sediment cores carries a high risk. There is too much uncertainty arising from the fact the 6+7Me-C17:0 concentrations only show very low correlation with FCA and the Nodularia and Aphanizomenon biomass data.

Answer: We are now giving both the 6Me-C17:0 and the 7Me-C17:0 data in Tables 3, 4, 5, and 7 (the table captions have been modified consequently). In each dataset, the 6Me-C17:0 to 7Me-C17:0 is around 0.2 (0.22 $\pm$ 0.02 for the trap sediments; 0.19 $\pm$ 0.03 for the short core sediments; 0.27 $\pm$ 0.07 for the long core sediments). These values are close to the 0.25 values for Nodularia as published in Bauersachs et al. Therefore, Nodularia is very likely the main source of 6- and 7-methylheptadecanes in the Bothnian Sea Holocene sediments. We have now added the following sentence in the Results: "Furthermore, Bauersachs et al. (2017) found that the 6Me-C17:0 to 7Me-C17:0 ratio was consistently around 0.25 in all four Nodularia spumigena strains. Similar values were found in the Baltic Sea sediments with 0.22 $\pm$ 0.02 (mean and standard deviation) in the trap sediments, 0.19 $\pm$ 0.03 in core MSM51-2/20 sediments, and 0.27 $\pm$ 0.07 in core POS435/10 sediments."

Additional points to address: Abstract Line 10 – Change "time history" to "history"

Answer: Done.

Line 12 – Change "trap sediments" to "sediment traps"

Answer: We kept here "trap sediments" as we are here talking about the sediments from the sediment traps. Line 19 – Remove "rather"

Answer: Done.

Introduction Lines 32 – 34 (and at other points in the manuscript). It does not read well to have a list of items, separated by commas but with multiple cases of "and" as the conjunction. For example, I would write the sentence as: "If diazotrophic cyanobacteria

occur in large blooms they contribute to nitrogen-eutrophication, where the massive export and decay of cyanobacterial biomass results in O2 consumption, leading to the spread of bottom water hypoxia and anoxia (Zillen and Conley, 2010; Feistel et al., 2016)."

Answer: We are grateful to the reviewer. The sentence has been changed as suggested and we have also verified for similar cases and modified them whenever possible.

Line 61 – Change to "considered well suited"

Answer: Done.

Discussion Line 244 – 246 – Strange wording. I would change to "The 6+7Me-C17:0 content is not significantly positively correlated to the FCA index (r = 0.08; p = 0.71; n =22), nor to the biomass of Nodularia spumigena (r = 0.10; p = 0.62; n = 26), nor to the biomass of Aphanizomenon sp. (r = -0.36; p = 0.07; n = 26)."

Answer: Changed.

Line 255 – Should be "which may be"

Answer: Changed.

Line 277 – Figure number missing.

Answer: A number has been added (Fig. 5C).

Please also note the supplement to this comment:
https://www.biogeosciences-discuss.net/bg-2019-455/bg-2019-455-AC1-supplement.pdf
* * *
[Figure]

| Sediment trap | | | | | | | | Monitoring | | | Satellite | |
|---|---|---|---|---|---|---|---|---|---|---|---|---|
| Date | $C_{17:0}$ (µg/m²/day) | 7Me-$C_{17:0}$ (µg/m²/day) | 6Me-$C_{17:0}$ (µg/m²/day) | 6+7Me-$C_{17:0}$ (µg/m²/day) | Aphanizomenon spp.* | Dolichospermum | Nodularia spp. | Date | Aphanizomenon spp. (mg/m³) | Nodularia spp. (mg/m³) | Date | FCA |
| 17.05.2010 | 529.8 | 29.9 | 5.4 | 35.3 | 1 | | | 06.05.2010 | 3.1 | -- | 02.06.2010 | 0.0 |
| 27.05.2010 | 28.4 | 7.0 | 1.6 | 8.7 | 1 | | | 17.05.2010 | 3.9 | -- | 07.06.2010 | 0.0 |
| 21.06.2010 | 5.2 | -- | -- | -- | -- | | | 03.06.2010 | 16.3 | -- | 12.06.2010 | 0.0 |
| 11.07.2010 | 0.0 | -- | -- | -- | -- | | | 01.07.2010 | 46.2 | 9.7 | 17.06.2010 | 0.0 |
| 16.07.2010 | 136.2 | 108.9 | 19.5 | 128.4 | 3 | 3 | 2 | 22.07.2010 | 42.1 | 95.3 | 22.06.2010 | 0.0 |
| 26.07.2010 | 2.9 | 2.9 | 0.6 | 3.4 | 3 | 3 | 2 | 19.08.2010 | 25.7 | 43.2 | 27.06.2010 | 0.0 |
| 05.08.2010 | 48.3 | 9.7 | 0.0 | 9.7 | 2 | 2 | 1 | 16.09.2010 | 6.6 | -- | 02.07.2010 | 0.2 |
| 15.08.2010 | 84.3 | 164.9 | 33.3 | 198.2 | 2 | 2 | 2 | 04.10.2010 | 36.0 | -- | 07.07.2010 | 5.3 |
| 04.09.2010 | 68.4 | 43.4 | 10.0 | 53.4 | 2 | 1 | 1 | 11.11.2010 | 4.6 | -- | 12.07.2010 | 38.4 |
| 24.09.2010 | 9.2 | -- | -- | -- | 1 | | | 16.11.2010 | 14.2 | -- | 17.07.2010 | 21.5 |
| 04.10.2010 | 164.3 | 38.3 | 9.7 | 48.1 | 3 | 1 | 1 | 13.01.2011 | -- | -- | 22.07.2010 | 21.4 |
| 24.10.2010 | 0.9 | -- | -- | -- | 2 | 1 | | 07.02.2011 | 0.4 | -- | 27.07.2010 | 15.8 |
| 25.11.2010 | 24.6 | -- | -- | -- | -- | | | 12.03.2011 | 0.1 | -- | 01.08.2010 | 0.6 |
| 01.12.2010 | 27.2 | -- | -- | -- | -- | | | 26.03.2011 | -- | -- | 06.08.2010 | 0.2 |
| 07.12.2010 | 19.2 | -- | -- | -- | 1 | | | 10.04.2011 | -- | -- | 11.08.2010 | 0.0 |
| 13.12.2010 | 57.6 | -- | -- | -- | 1 | | | 14.05.2011 | 19.8 | -- | 16.08.2010 | 0.6 |
| 25.12.2010 | 14.6 | -- | -- | -- | 2 | | | | | | 21.08.2010 | 0.0 |
| 09.01.2011 | 0.0 | -- | -- | -- | 1 | | | | | | 26.08.2010 | 1.1 |
| | | | | | | | | | | | 30.08.2010 | 0.0 |

\* 1: present; 2: abundant; 3: highly abundant

**Fig. 1.** Table for reviewer #1

**Supplement:**

**Supplementary material**

**Table S3.** Sediment trap, monitoring, and satellite data from the Eastern Gotland Basin.

| | Sediment trap | | | | | Monitoring | | | Satellite | |
|---|---|---|---|---|---|---|---|---|---|---|
| Date | 7Me-$C_{17:0}$ (µg/m$^2$/day) | 6Me-$C_{17:0}$ (µg/m$^2$/day) | 6+7Me-$C_{17:0}$ (µg/m$^2$/day) | *Aphanizomenon* spp.* | *Nodularia* spp.* | Date | *Aphanizomenon* spp. (mg/m$^3$) | *Nodularia* spp. (mg/m$^3$) | Date | FCA |
| 17.05.2010 | 29.9 | 5.4 | 35.3 | 1 | | 06.05.2010 | 3.1 | -- | 02.06.2010 | 0.0 |
| 27.05.2010 | 7.0 | 1.6 | 8.7 | 1 | | 17.05.2010 | 3.9 | -- | 07.06.2010 | 0.0 |
| 21.06.2010 | -- | -- | -- | -- | | 03.06.2010 | 16.3 | -- | 12.06.2010 | 0.0 |
| 11.07.2010 | -- | -- | -- | -- | | 01.07.2010 | 46.2 | 9.7 | 17.06.2010 | 0.0 |
| 16.07.2010 | 104.9 | 23.5 | 128.4 | 3 | 2 | 22.07.2010 | 42.1 | 95.3 | 22.06.2010 | 0.0 |
| 26.07.2010 | 2.9 | 0.6 | 3.4 | 3 | 2 | 19.08.2010 | 25.7 | 43.2 | 27.06.2010 | 0.0 |
| 05.08.2010 | 7.8 | 1.9 | 9.7 | 2 | 1 | 16.09.2010 | 6.6 | -- | 02.07.2010 | 0.2 |
| 15.08.2010 | 164.9 | 33.3 | 198.2 | 2 | 2 | 04.10.2010 | 36.0 | -- | 07.07.2010 | 5.3 |
| 04.09.2010 | 43.4 | 10.0 | 53.4 | 2 | 1 | 11.11.2010 | 4.6 | -- | 12.07.2010 | 38.4 |
| 24.09.2010 | -- | -- | -- | 1 | | 16.11.2010 | 14.2 | -- | 17.07.2010 | 21.5 |
| 04.10.2010 | 38.3 | 9.7 | 48.1 | 3 | 1 | 13.01.2011 | -- | -- | 22.07.2010 | 21.4 |
| 24.10.2010 | -- | -- | -- | 2 | | 07.02.2011 | 0.4 | -- | 27.07.2010 | 15.8 |
| 25.11.2010 | -- | -- | -- | -- | | 12.03.2011 | 0.1 | -- | 01.08.2010 | 0.6 |
| 01.12.2010 | -- | -- | -- | -- | | 26.03.2011 | -- | -- | 06.08.2010 | 0.2 |
| 07.12.2010 | -- | -- | -- | 1 | | 10.04.2011 | -- | -- | 11.08.2010 | 0.0 |
| 13.12.2010 | -- | -- | -- | 1 | | 14.05.2011 | 19.8 | -- | 16.08.2010 | 0.6 |
| 25.12.2010 | -- | -- | -- | 2 | | | | | 21.08.2010 | 0.0 |
| 09.01.2011 | -- | -- | -- | 1 | | | | | 26.08.2010 | 1.1 |
| | | | | | | | | | 30.08.2010 | 0.0 |

* 1: present; 2: abundant; 3: highly abundant

**Table S4.** Data from core MSM51-2/20, monitoring stations (summer and annual means), and satellite imagery (summer) for the period 1983-2016.

| Year | 7Me-$C_{17:0}$ (µg/gTOC) | 6Me-$C_{17:0}$ (µg/gTOC) | 6+7Me-$C_{17:0}$ (µg/gTOC) | Summer (July-August) | | Annual | | FCA |
|---|---|---|---|---|---|---|---|---|
| | | | | *Aphanizomenon* spp. (mg/m$^3$) | *Nodularia* spp. (mg/m$^3$) | *Aphanizomenon* spp. (mg/m$^3$) | *Nodularia* spp. (mg/m$^3$) | |
| 2016 | -- | | -- | 33.6 | 121.6 | 15.0 | 24.4 | 14.3 |
| 2015 | 0.96 | 0.23 | 1.19 | 120.5 | 66.0 | 51.6 | 17.1 | 19.2 |
| 2014 | -- | -- | -- | 75.6 | 161.6 | 21.0 | 38.5 | 43.5 |
| 2013 | -- | -- | -- | 266.0 | 37.1 | 97.3 | 10.3 | 6.0 |
| 2012 | 1.69 | 0.29 | 1.98 | 40.1 | 38.6 | 26.1 | 23.1 | 11.0 |
| 2011 | -- | -- | -- | 66.8 | 89.0 | 25.3 | 28.0 | 18.6 |
| 2010 | 1.05 | 0.18 | 1.22 | 36.3 | 57.3 | 15.0 | 16.0 | 11.0 |
| 2009 | -- | -- | -- | 186.3 | 48.9 | 59.9 | 14.3 | 5.7 |
| 2008 | 1.06 | 0.20 | 1.26 | 85.8 | 149.1 | 46.3 | 41.5 | 47.2 |
| 2007 | -- | -- | -- | 122.5 | 44.5 | 32.3 | 8.9 | 7.2 |
| 2006 | -- | -- | -- | 12.2 | 7.3 | 7.2 | 1.5 | 14.0 |
| 2005 | 1.03 | 0.23 | 1.26 | 145.0 | 335.4 | 48.7 | 84.6 | 44.8 |
| 2004 | -- | -- | -- | 73.2 | 256.8 | 33.6 | 77.5 | 9.7 |
| 2003 | 0.64 | 0.12 | 0.76 | 34.6 | 58.8 | 14.6 | 16.4 | 41.9 |
| 2002 | 0.29 | 0.05 | 0.34 | 109.6 | 177.6 | 38.6 | 49.4 | 17.6 |
| 2001 | 0.20 | 0.05 | 0.25 | 121.1 | 39.3 | 39.8 | 10.9 | 4.7 |
| 2000 | 0.46 | 0.07 | 0.52 | 25.2 | 113.6 | 12.3 | 28.4 | 25.7 |
| 1999 | 1.34 | 0.25 | 1.58 | 56.5 | 192.8 | 28.4 | 48.3 | 37.3 |
| 1998 | 0.63 | 0.11 | 0.74 | 52.0 | 41.4 | 26.4 | 10.7 | 7.7 |
| 1997 | 0.38 | 0.07 | 0.45 | 239.8 | 56.0 | 75.2 | 14.2 | 11.2 |
| 1996 | 0.02 | *n.d.* | 0.02 | 47.1 | 69.7 | 39.4 | 27.8 | 3.1 |
| 1995 | 0.05 | *n.d.* | 0.05 | 104.1 | 60.8 | 39.5 | 14.3 | 5.1 |
| 1994 | 0.08 | *n.d.* | 0.08 | 319.1 | 228.4 | 97.7 | 65.3 | 24.4 |
| 1993 | 0.22 | 0.06 | 0.28 | 141.6 | 84.0 | 47.9 | 25.1 | 6.7 |
| 1992 | 0.40 | 0.09 | 0.50 | 69.3 | 2148.8 | 20.4 | 538.0 | 17.9 |
| 1991 | 0.45 | 0.11 | 0.56 | 283.8 | 87.3 | 121.6 | 37.4 | 19.6 |
| 1990 | 0.53 | 0.12 | 0.65 | -- | -- | 33.9 | 7.5 | 1.6 |
| 1989 | 1.12 | 0.21 | 1.33 | -- | -- | 7.1 | 0.2 | 2.8 |
| 1988 | 1.23 | 0.22 | 1.44 | 70.5 | 19.5 | 6.9 | 0.0 | 3.3 |
| 1987 | 1.44 | 0.29 | 1.73 | 24.4 | 19.1 | 6.2 | 24.3 | 0.3 |
| 1986 | 0.88 | 0.16 | 1.04 | 41.2 | 3097.9 | 12.0 | 886.2 | -- |
| 1985 | 0.99 | 0.18 | 1.18 | 105.3 | 203.3 | 45.2 | 110.5 | -- |
| 1984 | 0.13 | *n.d.* | 0.13 | 46.3 | 16.3 | 17.5 | 5.4 | -- |
| 1983 | 0.15 | *n.d.* | 0.15 | 117.0 | 0.0 | 45.7 | 0.0 | -- |

**Table S5.** Core MSM51-2/20 (7Me-C$_{17:0}$, 6Me-C$_{17:0}$, 6+7Me-C$_{17:0}$ and TOC), HadISST1 (Rayner et al., 2003), AMO (Enfield et al., 2001), and NAO (Hurrell, 1995; Jones et al., 1997) data since 1860.

| Depth (cm) | Year | 7Me-C$_{17:0}$ (μg/gTOC) | 6Me-C$_{17:0}$ (μg/gTOC) | 6+7Me-C$_{17:0}$ (μg/gTOC) | TOC (%) | AMO | winter (DJFM) NAO | summer (J-A) HadISST1 (°C) |
|---|---|---|---|---|---|---|---|---|
| 0.25 | 2015 | 0.96 | 0.23 | 1.19 | 15.9 | 0.10 | 8.16 | -- |
| -- | 2014 | -- | -- | -- | -- | 0.09 | 8.21 | -- |
| -- | 2013 | -- | -- | -- | -- | 0.15 | -2.33 | -- |
| 0.75 | 2012 | 1.69 | 0.29 | 1.98 | 14.6 | 0.20 | 8.23 | -- |
| -- | 2011 | -- | -- | -- | -- | 0.09 | -3.65 | -- |
| 1.25 | 2010 | 1.05 | 0.18 | 1.22 | 15.2 | 0.34 | -10.15 | -- |
| -- | 2009 | -- | -- | -- | -- | 0.02 | -1.25 | -- |
| 1.75 | 2008 | 1.06 | 0.20 | 1.26 | 14.4 | 0.12 | 5.47 | 17.6 |
| -- | 2007 | -- | -- | -- | -- | 0.13 | 7.30 | 16.6 |
| -- | 2006 | -- | -- | -- | -- | 0.25 | -0.79 | 18.7 |
| 2.25 | 2005 | 1.03 | 0.23 | 1.26 | 16.0 | 0.28 | -0.45 | 17.4 |
| -- | 2004 | -- | -- | -- | -- | 0.19 | -0.81 | 17.1 |
| 2.75 | 2003 | 0.64 | 0.12 | 0.76 | 14.6 | 0.22 | 1.59 | 18.0 |
| 3.50 | 2002 | 0.29 | 0.05 | 0.34 | 11.3 | 0.05 | 3.16 | 18.3 |
| 4.25 | 2001 | 0.20 | 0.05 | 0.25 | 10.9 | 0.10 | -2.00 | 18.0 |
| 4.75 | 2000 | 0.46 | 0.07 | 0.52 | 10.8 | 0.01 | 7.39 | 16.0 |
| 5.50 | 1999 | 1.34 | 0.25 | 1.58 | 10.6 | 0.10 | 3.93 | 18.0 |
| 6.25 | 1998 | 0.63 | 0.11 | 0.74 | 11.3 | 0.36 | 3.20 | 15.5 |
| 7.00 | 1997 | 0.38 | 0.07 | 0.45 | 10.6 | 0.04 | 0.70 | 18.6 |
| 7.75 | 1996 | 0.02 | *n.d.* | 0.02 | 12.1 | -0.07 | -9.29 | 15.9 |
| 8.25 | 1995 | 0.05 | *n.d.* | 0.05 | 12.5 | 0.12 | 9.75 | 16.4 |
| 8.75 | 1994 | 0.08 | *n.d.* | 0.08 | 9.6 | -0.19 | 7.20 | 17.8 |
| 9.25 | 1993 | 0.22 | 0.06 | 0.28 | 9.2 | -0.23 | 5.70 | 15.4 |
| 9.75 | 1992 | 0.40 | 0.09 | 0.50 | 11.1 | -0.23 | 6.72 | 16.4 |
| 10.25 | 1991 | 0.45 | 0.11 | 0.56 | 11.7 | -0.15 | 0.82 | 16.8 |
| 10.75 | 1990 | 0.53 | 0.12 | 0.65 | 11.7 | -0.05 | 9.49 | 16.3 |
| 11.25 | 1989 | 1.12 | 0.21 | 1.33 | 9.9 | -0.10 | 11.44 | 16.6 |
| 11.75 | 1988 | 1.23 | 0.22 | 1.44 | 9.6 | -0.02 | 0.39 | 16.9 |
| 12.25 | 1987 | 1.44 | 0.29 | 1.73 | 9.8 | 0.05 | 1.35 | 13.5 |
| 12.75 | 1986 | 0.88 | 0.16 | 1.04 | 10.6 | -0.29 | -0.13 | 15.9 |
| 13.25 | 1985 | 0.99 | 0.18 | 1.18 | 10.7 | -0.28 | -1.52 | 15.6 |
| 13.75 | 1984 | 0.13 | *n.d.* | 0.13 | 9.6 | -0.22 | 2.97 | 16.6 |
| 14.25 | 1983 | 0.15 | *n.d.* | 0.15 | 10.4 | -0.09 | 8.00 | 16.2 |
| 14.75 | 1982 | 0.13 | *n.d.* | 0.13 | 9.6 | -0.23 | 0.99 | 16.6 |
| 15.25 | 1981 | 0.08 | *n.d.* | 0.15 | 8.9 | -0.09 | 3.60 | 15.1 |
| -- | 1980 | -- | -- | -- | -- | -0.03 | 0.29 | 16.1 |
| 15.75 | 1979 | 0.16 | *n.d.* | 0.16 | 7.6 | -0.13 | -5.38 | 15.4 |
| 16.25 | 1978 | 0.08 | *n.d.* | 0.08 | 8.3 | -0.19 | 1.32 | 15.1 |
| 16.75 | 1977 | 0.06 | *n.d.* | 0.06 | 8.9 | -0.20 | -4.38 | 14.8 |
| 17.25 | 1976 | 0.06 | *n.d.* | 0.06 | 8.7 | -0.38 | 2.34 | 15.8 |
| 17.75 | 1975 | 0.30 | 0.11 | 0.41 | 10.2 | -0.31 | 4.63 | 17.1 |
| -- | 1974 | -- | -- | -- | -- | -0.43 | 1.97 | 15.7 |
| 18.25 | 1973 | 0.68 | 0.15 | 0.83 | 8.4 | -0.23 | 5.74 | 17.8 |
| 18.75 | 1972 | 0.47 | 0.07 | 0.54 | 8.5 | -0.37 | 0.30 | 17.0 |
| 19.25 | 1971 | 0.44 | *n.d.* | 0.38 | 8.2 | -0.32 | -2.56 | 15.2 |
| -- | 1970 | -- | -- | -- | -- | -0.12 | -2.10 | 15.6 |
| 19.75 | 1969 | 0.51 | 0.09 | 0.60 | 9.0 | 0.00 | -8.34 | 17.3 |
| 20.25 | 1968 | 0.18 | *n.d.* | 0.18 | 8.4 | -0.18 | -0.07 | 16.3 |
| -- | 1967 | -- | -- | -- | -- | -0.11 | 6.42 | 16.1 |
| 20.75 | 1966 | 0.10 | *n.d.* | 0.10 | 8.6 | -0.01 | 0.90 | 15.7 |
| 21.25 | 1965 | 0.20 | *n.d.* | 0.20 | 7.9 | -0.17 | -4.03 | 14.1 |
| 21.75 | 1964 | 0.21 | *n.d.* | 0.21 | 8.1 | -0.11 | -3.06 | 14.7 |
| -- | 1963 | -- | -- | -- | -- | -0.01 | -3.86 | 15.7 |
| 22.25 | 1962 | 0.14 | *n.d.* | 0.14 | 7.6 | 0.06 | -2.29 | 13.9 |
| 22.75 | 1961 | 0.60 | 0.14 | 0.74 | 9.8 | 0.09 | 8.06 | 15.1 |
| 23.25 | 1960 | 0.82 | 0.12 | 0.94 | 9.4 | 0.23 | -0.59 | 15.6 |
| 23.75 | 1959 | 0.26 | *n.d.* | 0.26 | 6.8 | 0.04 | 1.43 | 16.9 |
| 24.25 | 1958 | 0.31 | *n.d.* | 0.31 | 6.5 | 0.21 | -1.06 | 14.6 |
| 24.75 | 1957 | 0.20 | *n.d.* | 0.20 | 7.0 | 0.03 | 6.60 | 15.4 |
| 25.25 | 1956 | 0.11 | *n.d.* | 0.11 | 4.9 | -0.03 | -3.20 | 14.8 |
| 25.75 | 1955 | 1.00 | 0.24 | 1.24 | 3.0 | 0.18 | -4.84 | 16.4 |
| -- | 1954 | -- | -- | -- | -- | 0.03 | 0.53 | 15.7 |
| 26.25 | 1953 | 0.71 | 0.15 | 0.86 | 3.3 | 0.26 | 1.46 | 16.4 |
| -- | 1952 | -- | -- | -- | -- | 0.29 | 2.47 | 15.5 |
| 26.75 | 1951 | 0.48 | *n.d.* | 0.48 | 3.3 | 0.20 | -2.14 | 15.3 |
| -- | 1950 | -- | -- | -- | -- | -0.02 | 4.79 | 16.9 |
| 27.25 | 1949 | 0.41 | *n.d.* | 0.41 | 2.6 | 0.09 | 5.64 | 15.8 |
| -- | 1948 | -- | -- | -- | -- | 0.01 | 4.95 | 15.3 |
| 27.75 | 1947 | 0.35 | *n.d.* | 0.35 | 2.7 | -0.09 | -4.27 | 16.4 |
| -- | 1946 | -- | -- | -- | -- | 0.01 | 1.76 | 16.0 |
| 28.25 | 1945 | 0.41 | *n.d.* | 0.41 | 3.1 | 0.21 | 5.32 | 16.7 |
| 28.75 | 1944 | 0.52 | 0.10 | 0.62 | 3.1 | 0.34 | 0.83 | 16.1 |
| -- | 1943 | -- | -- | -- | -- | 0.03 | 5.45 | 15.7 |
| 29.25 | 1942 | 0.57 | 0.13 | 0.70 | 3.0 | 0.18 | -1.83 | 14.9 |
| -- | 1941 | -- | -- | -- | -- | 0.17 | -3.75 | 14.7 |
| 29.75 | 1940 | 0.71 | 0.17 | 0.88 | 3.0 | -0.03 | -4.99 | 15.7 |
| -- | 1939 | -- | -- | -- | -- | 0.11 | 1.80 | 17.5 |
| -- | 1938 | -- | -- | -- | -- | 0.24 | 5.28 | 17.2 |

| | | | | | | | | |
|---|---|---|---|---|---|---|---|---|
| 30.50 | 1937 | 0.95 | 0.24 | 1.19 | 2.5 | 0.29 | 4.23 | 18.8 |
| -- | 1936 | -- | -- | -- | -- | 0.15 | -6.12 | 17.8 |
| -- | 1935 | -- | -- | -- | -- | 0.02 | 3.91 | 16.3 |
| -- | 1934 | -- | -- | -- | -- | -0.01 | 2.22 | 16.9 |
| -- | 1933 | -- | -- | -- | -- | 0.19 | 2.08 | 16.6 |
| -- | 1932 | -- | -- | -- | -- | 0.23 | -0.64 | 17.7 |
| 31.50 | 1931 | 0.87 | 0.15 | 1.02 | 2.5 | 0.18 | 0.23 | 14.8 |
| -- | 1930 | -- | -- | -- | -- | 0.01 | 4.01 | 16.8 |
| -- | 1929 | -- | -- | -- | -- | -0.11 | -1.03 | 14.2 |
| -- | 1928 | -- | -- | -- | -- | -0.01 | 2.48 | 13.9 |
| -- | 1927 | -- | -- | -- | -- | 0.11 | 5.31 | 17.1 |
| -- | 1926 | -- | -- | -- | -- | 0.08 | 3.38 | 16.8 |
| -- | 1925 | -- | -- | -- | -- | -0.16 | 8.04 | 16.5 |
| -- | 1924 | -- | -- | -- | -- | -0.15 | -1.39 | 15.8 |
| -- | 1923 | -- | -- | -- | -- | -0.33 | 5.98 | 14.7 |
| 32.50 | 1922 | 0.39 | n.d. | 0.39 | 2.5 | -0.32 | 5.56 | 15.6 |
| -- | 1921 | -- | -- | -- | -- | -0.22 | 5.49 | 15.1 |
| -- | 1920 | -- | -- | -- | -- | -0.34 | 10.17 | 15.7 |
| -- | 1919 | -- | -- | -- | -- | -0.19 | 1.57 | 14.4 |
| -- | 1918 | -- | -- | -- | -- | -0.26 | 0.96 | 15.7 |
| -- | 1917 | -- | -- | -- | -- | -0.28 | -6.68 | 15.0 |
| -- | 1916 | -- | -- | -- | -- | -0.08 | 1.36 | 14.6 |
| -- | 1915 | -- | -- | -- | -- | 0.09 | 2.22 | 15.4 |
| -- | 1914 | -- | -- | -- | -- | -0.29 | 5.98 | 18.2 |
| 33.50 | 1913 | 0.16 | n.d. | 0.16 | 2.5 | -0.39 | 8.75 | 16.1 |
| -- | 1912 | -- | -- | -- | -- | -0.23 | 4.16 | 16.9 |
| -- | 1911 | -- | -- | -- | -- | -0.21 | 1.37 | 15.8 |
| -- | 1910 | -- | -- | -- | -- | -0.25 | 6.49 | 15.2 |
| -- | 1909 | -- | -- | -- | -- | -0.14 | 0.66 | 14.7 |
| -- | 1908 | -- | -- | -- | -- | -0.13 | 4.54 | 15.8 |
| -- | 1907 | -- | -- | -- | -- | -0.23 | 5.24 | 13.3 |
| -- | 1906 | -- | -- | -- | -- | -0.07 | 5.50 | 15.9 |
| -- | 1905 | -- | -- | -- | -- | -0.20 | 7.01 | 16.6 |
| 34.50 | 1904 | 0.09 | n.d. | 0.09 | 2.4 | -0.35 | 1.66 | 13.9 |
| -- | 1903 | -- | -- | -- | -- | -0.19 | 11.46 | 14.4 |
| -- | 1902 | -- | -- | -- | -- | -0.10 | -0.87 | 14.2 |
| -- | 1901 | -- | -- | -- | -- | 0.09 | 0.99 | 16.0 |
| -- | 1900 | -- | -- | -- | -- | 0.10 | -4.97 | 15.6 |
| -- | 1899 | -- | -- | -- | -- | 0.13 | 2.15 | 16.2 |
| 35.50 | 1898 | 0.04 | n.d. | 0.04 | 2.4 | 0.08 | 2.89 | 15.1 |
| -- | 1897 | -- | -- | -- | -- | 0.11 | 5.13 | 15.7 |
| -- | 1896 | -- | -- | -- | -- | 0.11 | 3.51 | 16.9 |
| -- | 1895 | -- | -- | -- | -- | -0.09 | -6.08 | 15.3 |
| -- | 1894 | -- | -- | -- | -- | -0.24 | 8.33 | 15.6 |
| 36.50 | 1893 | 0.08 | n.d. | 0.08 | 2.3 | 0.00 | -0.16 | 15.5 |
| -- | 1892 | -- | -- | -- | -- | -0.09 | -1.65 | 15.0 |
| -- | 1891 | -- | -- | -- | -- | 0.04 | -0.84 | 16.0 |
| -- | 1890 | -- | -- | -- | -- | -0.14 | 5.75 | 15.3 |
| 37.50 | 1889 | 0.04 | n.d. | 0.04 | 2.4 | 0.20 | 1.81 | 15.5 |
| -- | 1888 | -- | -- | -- | -- | 0.20 | -5.16 | 14.0 |
| -- | 1887 | -- | -- | -- | -- | 0.12 | 3.20 | 15.1 |
| -- | 1886 | -- | -- | -- | -- | 0.13 | -1.27 | 15.8 |
| -- | 1885 | -- | -- | -- | -- | -0.02 | 0.17 | 15.8 |
| 38.50 | 1884 | 0.04 | n.d. | 0.04 | 2.5 | -0.07 | 5.90 | 16.7 |
| -- | 1883 | -- | -- | -- | -- | -0.03 | 1.19 | 15.9 |
| -- | 1882 | -- | -- | -- | -- | -0.02 | 12.65 | 16.7 |
| -- | 1881 | -- | -- | -- | -- | 0.05 | -5.21 | 14.8 |
| 39.50 | 1880 | 0.12 | n.d. | 0.12 | 2.5 | 0.07 | 2.98 | 15.8 |
| -- | 1879 | -- | -- | -- | -- | 0.13 | -3.25 | 15.3 |
| -- | 1878 | -- | -- | -- | -- | 0.46 | 5.53 | 15.7 |
| -- | 1877 | -- | -- | -- | -- | 0.25 | 2.77 | 14.6 |
| 40.50 | 1876 | 0.09 | n.d. | 0.09 | 2.5 | -0.02 | 1.64 | 15.5 |
| -- | 1875 | -- | -- | -- | -- | 0.04 | -2.14 | 16.5 |
| -- | 1874 | -- | -- | -- | -- | -0.01 | 7.11 | 15.3 |
| -- | 1873 | -- | -- | -- | -- | 0.05 | 1.00 | 15.6 |
| -- | 1872 | -- | -- | -- | -- | 0.10 | 0.22 | 16.3 |
| 41.50 | 1871 | 0.12 | n.d. | 0.12 | 2.5 | 0.04 | 0.01 | 14.6 |
| -- | 1870 | -- | -- | -- | -- | 0.03 | -4.42 | 15.5 |
| -- | 1869 | -- | -- | -- | -- | 0.10 | 7.25 | -- |
| -- | 1868 | -- | -- | -- | -- | 0.16 | 7.80 | -- |
| 42.50 | 1867 | 0.21 | n.d. | 0.21 | 2.4 | 0.15 | 0.57 | -- |
| -- | 1866 | -- | -- | -- | -- | 0.21 | 3.23 | -- |
| -- | 1865 | -- | -- | -- | -- | 0.18 | -2.07 | -- |
| -- | 1864 | -- | -- | -- | -- | 0.07 | 2.15 | -- |
| -- | 1863 | -- | -- | -- | -- | -0.11 | 9.02 | -- |
| -- | 1862 | -- | -- | -- | -- | -0.19 | -3.86 | -- |
| -- | 1861 | -- | -- | -- | -- | 0.19 | 0.16 | -- |
| -- | 1860 | -- | -- | -- | -- | 0.09 | 1.63 | -- |

*n.d. = not detected*

**Table S7.** Core POS435/10 data.

| Depth (cm) | Age (yr cal. BP) | 7Me-$C_{17:0}$ (µg/gTOC) | 6Me-$C_{17:0}$ (µg/gTOC) | 6+7Me-$C_{17:0}$ (µg/gTOC) |
|---|---|---|---|---|
| 0.25 | -61.8 | 0.9 | 0.3 | 1.2 |
| 32.5 | -40.9 | 0.2 | 0.0 | 0.3 |
| 58.5 | -0.6 | 1.1 | 0.3 | 1.4 |
| 85 | 50.0 | 0.2 | 0.0 | 0.3 |
| 115 | 130.1 | 0.5 | 0.2 | 0.7 |
| 145 | 210.1 | 1.2 | 0.2 | 1.4 |
| 175 | 290.2 | 0.7 | 0.2 | 0.9 |
| 205 | 370.3 | 2.0 | 0.3 | 2.3 |
| 260 | 533.5 | 0.6 | 0.2 | 0.9 |
| 310 | 717.0 | 0.2 | 0.0 | 0.2 |
| 340 | 827.1 | 0.3 | 0.1 | 0.4 |
| 410 | 1083.9 | 0.5 | 0.1 | 0.6 |
| 465 | 1285.7 | 1.1 | 0.2 | 1.3 |
| 500 | 1414.2 | 0.3 | 0.1 | 0.4 |
| 520 | 1515.3 | 0.2 | 0.0 | 0.2 |
| 540 | 1617.2 | 1.0 | 0.3 | 1.3 |
| 560 | 1719.0 | 0.7 | 0.2 | 0.9 |
| 580 | 1820.9 | 1.2 | 0.3 | 1.5 |
| 600 | 1922.7 | 0.2 | 0.0 | 0.2 |
| 610 | 1977.4 | 0.4 | 0.1 | 0.5 |
| 620 | 2053.1 | 0.2 | 0.1 | 0.3 |
| 640 | 2204.7 | 1.6 | 0.4 | 1.9 |
| 650 | 2280.5 | 1.9 | 0.6 | 2.5 |
| 660 | 2466.8 | 2.6 | 0.6 | 3.2 |
| 670 | 2642.3 | 1.8 | 0.5 | 2.3 |
| 680 | 2817.8 | 0.2 | 0.1 | 0.3 |
| 707 | 3291.6 | 0.1 | 0.0 | 0.1 |
| 731 | 3733.3 | 0.1 | 0.0 | 0.1 |
| 739 | 3884.0 | 0.1 | 0.1 | 0.2 |
| 747 | 4034.7 | 0.7 | 0.2 | 0.9 |
| 759 | 4395.9 | 1.9 | 0.4 | 2.4 |
| 771 | 4824.8 | 0.3 | 0.1 | 0.4 |
| 783 | 5253.7 | 0.1 | 0.1 | 0.2 |
| 791 | 5539.7 | 0.1 | 0.1 | 0.2 |
| 803 | 5833.1 | 0.3 | 0.1 | 0.4 |
| 811 | 5922.0 | 0.1 | 0.0 | 0.2 |
| 817 | 5988.7 | 0.2 | 0.1 | 0.2 |
| 833 | 6166.4 | 9.5 | 3.0 | 12.5 |
| 839 | 6230.2 | 17.2 | 4.0 | 21.2 |
| 847 | 6296.1 | 61.3 | 15.6 | 76.9 |
| 863 | 6529.2 | 51.1 | 11.0 | 62.1 |
| 871 | 6662.6 | 218.2 | 62.7 | 280.9 |
| 879 | 6808.6 | 220.0 | 66.6 | 286.6 |
| 893.5 | 7080.8 | 0.2 | 0.0 | 0.3 |
| 905.5 | 7306.0 | 0.3 | 0.1 | 0.4 |

---

## Author Comment (AC2) · 7 Feb 2020

Referee #2 Tom Jilbert This manuscript presents several interesting datasets concerning the past abundance of diazotrophic cyanobacteria in the Baltic and Bothnian Seas on various timescales. The datasets are derived both from direct observations (water column- and sediment trap- monitoring of genera abundance, as well as satellite-based observations of bloom frequency) and from a new organic proxy in sediment trap and core samples, namely the abundance of mid-chain branched alkane (6+7Me-C17:0) lipids. The main goal of the study is test the applicability of these biomarkers for the

reconstruction of past diazotrophic cyanobacterial abundance, and indeed the authors present one such long sediment core record from the Bothnian Sea. The authors also use their biomarker data, along with instrumental and proxy-derived time series of climatic parameters, to investigate the potential climatic forcing of bloom occurrence on various timescales. The intrinsic value of the proxy seems to be high, and the paper is well written. However I have concerns over the authors' conclusions about the drivers of cyanobacterial bloom occurrence on various timescales, in particular their strong favoring of temperature over nutrient dynamics. Major comments 1. The conclusion stated several times in the manuscript, e.g. Line 323, "This record suggests that cyanobacterial blooms have not increased due to anthropogenic nutrient loading" is too bold considering the data presented in the study. Most researchers would agree that cyanobacterial bloom occurrence during recent decades is influenced by both temperature and nutrient dynamics, so without a strong piece of evidence to refute one or other factor, I suggest to moderate the wording of these sections. I also suggest that the authors should add the 20th century nutrient loading time series to Figure 5, in order for the reader to see how this compares with the other data presented. Some further considerations related to this: - The principal reasoning for stating that blooms respond more to temperature than nutrient loading is the "early onset" of blooms in the 20th century as implied by the peak in 6+7Me-C17:0 lipids in the period 1920-1940 in the Gotland Basin core. However, monitoring data show that phosphorus loading during this period increased by some 20% with respect to the period 1900-1920 (see below). This increase of 20% is greater in absolute terms (and certainly less noisy) than the SST increase over the same period (approx. 1°C +/- 2°C, see below). Of course, in the case of both nutrient loading and temperature, the response of cyanobacterial blooms is expected to be non-linear, e.g. a threshold-type response, related to the fact that these organisms are competing for resources within an ecosystem, and above certain thresholds of certain environmental variables may gain significant competitive advantage over other primary producers. Hence the difficulty in making direct linear correlation analyses with time series of those environmental variables. In summary,

I would like to see a more balanced acknowledgement that both nutrient loading and temperature may have influenced bloom occurrence during this period, and that these responses are likely strongly non-linear.

Answer: In agreement with Mr. Jilbert's comment, we have now included Gustafsson et al. data of riverine phosphorus input to the Baltic Sea in Figure 5. We have added Gustafsson et al. (2012) to the reference list. The caption of Figure 5 has been modified consequently. We have also moderated our wording and suggest now that both temperature and nutrient inputs likely influenced cyanobacterial blooms. We have added a few sentences in the manuscript: - Abstract: "While the early increase in cyanobacteria may be related to a small increase in phosphorus loading, decadal to multi-decadal fluctuations are likely related to variability in the Baltic Sea surface temperature and, ultimately, to the Atlantic Multidecadal Oscillation". - Section 4.3: "However, the small increase (ca. 16 %) of riverine phosphorus input to the Baltic Sea may have partly triggered the early increase in cyanobacteria abundance in the 1920s (Fig. 5B)." - Conclusions: "This record suggests that anthropogenic nutrient loading is likely not the main trigger for cyanobacterial blooms, but may have favoured an early increase in the 1920s. Cyanobacterial biomass fluctuation seems to be at least partly related to sea surface temperature changes and the AMO climate mode at a decadal to multi-decadal timescale over the last 140 years." Concerning the possible non-linear response of cyanobacterial occurrence to environmental parameters, we have now added this sentence at the end of Section 4.3: "However, it has to be considered that the response of cyanobacterial blooms to environmental variables such as nutrient loading and temperature are likely non-linear, that may explain the relatively low correlations. Indeed, a very recent study shows that cyanobacterial blooms are highly correlated to environmental variables at a decadal timescale when considering a set of biogeochemical variables related to the amount of phosphorus and hypoxia in bottom layers, as well as surface water temperature (Kahru et al., 2020)." Note that the Kahru et al. (2020) article has been added to the reference list and is attached.

2. I have a similar concern with the interpretation of the long sediment core record in Figure 6, although now we are discussing natural rather than anthropogenically-impacted nutrient cycling. The authors acknowledge in the text that phosphorus regeneration played an important role in sustaining blooms during the HTM in the Bothnian Sea, as we showed in our earlier study (Jilbert et al., 2015). However I would also like to see a statement acknowledging that declining P availability was likely the main factor in the steep decline in blooms from 6500 yr. B.P., which is a dominant feature of this record (we interpreted this as due to the shoaling of the Åland Sea sills). The temperature records from the Swedish lakes presented here support the concept of warm conditions favoring blooms during the HTM, but for example, 4500 yr. B.P. shows a similar temperature to 6500 yr. B.P., yet the bloom intensity is orders of magnitude lower as shown by the log-scale of 6+7Me-C17:0. This requires another controlling factor, ie. availability of P.

Answer: In agreement with Mr. Jilbert, we have now mentioned that the shoaling of the Åland Sea sills may have declined P availability, and thus diminished the cyanobacterial blooms after ca. 6300 yrs BP. We have added this sentence in Section 4.4: "However, as suggested by Jilbert et al. (2015), the shoaling of the Åland Sea sills (Fig. 1) and the resulting decline in P availability may have been a major factor explaining the decline in cyanobacterial blooms from ca. 6300 years cal. BP." Figure 1 and its caption have been slightly modified to add the location of the Åland Sea.

Minor comments: Line 96: Replace 'bloom' with 'blooms'

Answer: Done.

Line 97: One could reasonably ask why a core from the Bothnian Sea is investigated and not from the same location as the short cores and sediment trap series Biomarker and SST time series presented in the study.

Answer: The aim here was only to illustrate the fact that the 6+7Me-C17:0 cyanobacterial biomarker is working for the complete Holocene period. We have chosen this core

from the Bothnian Sea because of its excellent age model and for sample availability. We are now working on a multi-proxy study including the 6+7Me-C17:0 cyanobacterial biomarker on this core with a higher temporal resolution.

Line 106: Give more detail on the coring device

Answer: This sentence has been added: "The short sediment core MSM51-2/20 was retrieved with a device keeping the sediment-water interface undisturbed (multi-corer). The core was sampled every 0.5 to 1 cm. The sediment samples were frozen-dried and homogenized (n = 73). The long sediment core POS435/10 retrieved with a gravity corer (Häusler et al., 2017) was sampled every 10-20 cm". In our opinion, these coring devices are nowadays relatively standard in the field and a more detailed description of their functioning is not required here.

Line 111: Methodology for estimating TOC needs more detail. Is one of these instruments able to isolate and measure inorganic carbon from a bulk sample?

Answer: We have now given more details on the method used to estimate TOC: "Total organic carbon (TOC) was calculated by the subtraction of total inorganic carbon (TIC) from total carbon (TC) values. TC was analysed by means of an EA1110 CHN (CE-instruments). TIC was determined by means of a TIC module connected with a Multi EA 2000 CS (Analytik, Jena) elemental analyser, involving the acidic removal of carbonates from sediment samples and analysis of the $CO_2$ released in a carrier gas stream (Leipe et al., 2011)."

Line 255: Replace 'what' with 'which'

Answer: Done.

Please also note the supplement to this comment:
https://www.biogeosciences-discuss.net/bg-2019-455/bg-2019-455-AC2-supplement.pdf

[Figure]

Figure 5

**Fig. 1.** Revised Figure 5

[Figure]

**Fig. 2.** Revised Figure 1

**Supplement:**

Harmful Algae 92 (2020) 101739

Contents lists available at ScienceDirect

**Harmful Algae**

journal homepage: www.elsevier.com/locate/hal

[Figure]

**Cyanobacterial blooms in the Baltic Sea: Correlations with environmental factors**

[Figure]

Mati Kahru[a,*], Ragnar Elmgren[b], Jérôme Kaiser[c], Norbert Wasmund[c], Oleg Savchuk[d]

[a] *Scripps Institution of Oceanography, University of California San Diego, La Jolla, CA 92093-0218, USA*
[b] *Department of Ecology, Environment and Plant Sciences, Stockholm University, SE-10691 Stockholm, Sweden*
[c] *Leibniz Institute for Baltic Sea Research (IOW), 18119 Rostock-Warnemünde, Germany*
[d] *Baltic Nest Institute, Stockholm University, SE-10691 Stockholm, Sweden*

**ARTICLE INFO**

*Keywords:*
Cyanobacteria
Surface accumulations
Baltic Sea
Satellite
Excess phosphorus
Solar flux
Sea-surface temperature

**ABSTRACT**

Massive cyanobacteria blooms occur almost every summer in the Baltic Sea but the capability to quantitatively predict their extent and intensity is poorly developed. Here we analyse statistical relationships between multi-decadal satellite-derived time series of the frequency of cyanobacteria surface accumulations (FCA) in the central Baltic Sea Proper and a suite of environmental variables. Over the decadal scale (~5-20 years) FCA was highly correlated ($R^2 \sim 0.69$) with a set of biogeochemical variables related to the amount of phosphorus and hypoxia in bottom layers. Water temperature in the surface layer was also positively correlated with FCA at the decadal scale. In contrast, the inter-annual variations in FCA had no correlation with the biogeochemical variables. Instead, significant correlations were found with the solar shortwave direct flux in July and the sea-surface temperature, also in July. It thus appears that it is not possible to predict inter-annual fluctuations in cyano-bacteria blooms from water chemistry. Moreover, environmental variables could only explain about 45% of the inter-annual variability in FCA, probably because year-to-year variations in FCA are significantly influenced by biological interactions.

**1. Introduction**

Nitrogen-fixing cyanobacteria have been an important component of the Baltic Sea ecosystem for millennia (Bianchi et al., 2000; Funkey et al., 2014) and contribute substantially to ecosystem productivity when inorganic nitrogen is in short supply (Larsson et al., 2001; Karlson et al., 2015). As many cyanobacteria are toxic and form harmful algal blooms, there has been a lot of effort to identify environmental factors that favour such blooms. Numerous publications (Niemi, 1979; Kahru et al., 1994; Wasmund, 1997; Paerl and Huisman, 2008, 2009; Paerl et al., 2011) have pointed out environmental conditions that enhance the competitive advantage and growth of cyanobacteria such as high inorganic phosphorus to nitrogen ratio, high water temperature, high solar irradiance and low winds. However, these conditions are not sufficient to quantitatively predict the concentration and the extent of cyanobacterial blooms. As a first step towards such predictions, we here analyse statistical relationships between a unique multi-decadal satellite-detected time series of near-surface cyanobacteria concentrations in the Baltic Sea (Kahru and Elmgren, 2014, extended to 2018) and various environmental variables.

The spatial patterns of cyanobacteria accumulations are extremely patchy with the characteristic multi-scale stripes and swirls (e.g. Kahru et al., 1994; Kutser, 2004) that makes their sampling with conventional water samplers highly unreliable. We used a satellite-derived time series of over three decades that is made possible by the tendency by one of the co-dominant cyanobacteria species in the open Baltic Sea, *Nodularia spumigena*, to form dense surface accumulations. The other co-dominant species, *Aphanizomenon* sp., is typically distributed deeper in the water column (Hajdu et al., 2007; Rolff et al., 2007) and there-fore not specifically detected in this time series. Since *Aphanizomenon* in the Baltic Sea is non-toxic and contributes little to the surface blooms, the blooms registered by satellites provide valid measurements of the frequency and extent of cyanobacterial blooms of main societal con-cern. The satellite data used here are based on a pixel size of $\sim 1\,km^2$, which is above the scale of typical stripes and swirls seen from ships. However, the sub-pixel structures are integrated into the pixel values that are well correlated with higher resolution measurements, e.g. from continuous in-water measurements along ship tracks (Kahru and Elmgren, 2014).

While statistical relationships do not prove causation, they provide
* * *
* Corresponding author.
  *E-mail address:* mkahru@ucsd.edu (M. Kahru).

https://doi.org/10.1016/j.hal.2019.101739
Received 22 August 2019; Received in revised form 7 October 2019; Accepted 30 December 2019
1568-9883/ © 2019 Published by Elsevier B.V.

clues to finding the environmental conditions that regulate cyano-bacterial blooms in the Baltic Sea, and could help in modelling efforts to predict the likelihood of extensive blooms in the future. We use a statistical method called partial least squares (PLS) regression, a technique that combines the features of principal component analysis and multiple linear regression (CenterSpace, 2016). In the terminology of PLS we try to predict one or more variables from combinations of a set of variables, some of which may be highly correlated. Whereas the principal components regression computes the factor scores using the covariance structure between predictor variables, PLS regression computes factor scores from the covariance structure between the predictor and the response variables. The term prediction is used here not as a forecast into the future but as a way to statistically fit the measured time series with a modelled time series using a suite of input variables.

**2. Methods**

**2.1. Frequency of Cyanobacteria accumulations (FCA)**

To characterize the frequency and extent of cyanobacteria blooms we have developed an index (Kahru et al., 2007; Kahru and Elmgren, 2014) called the frequency of cyanobacteria accumulations (FCA). FCA is the ratio of the number of days when cyanobacteria accumulations were detected to the number of days with unobstructed (cloud-free) satellite views of the sea surface, calculated for each pixel, and is not directly dependent on the number of available images. As satellite detection of cyanobacteria accumulations is limited to periods of clear skies and availability of near-noon satellite overpasses, FCA normalizes the number of detections to the number of observations. Even though normalized, FCA values are less reliable if the number of available images is small. As most environmental datasets used here (see below) start in 1987, we use FCA time series from 1987 to 2018.

Cyanobacteria blooms in the Baltic Sea often start already in June but the bulk of the accumulations occurs during the months of July and August. We therefore use FCA calculated over the 2-month period of July-August as the annual estimate of the strength of the blooms.

In past publications (Kahru et al., 2007; Kahru and Elmgren, 2014), we used FCA calculated over an area that covered most of the Baltic Sea including the gulfs of Bothnia, Finland and Riga. As near-coast turbidity and resuspended sediments in shallow areas can interfere with our method of cyanobacteria detection, we restrict our analysis here to the central Baltic Proper deeper than 15 m (Fig. 1). In other aspects the FCA time series follows the methods of Kahru and Elmgren (2014) and was extended to 2018 by using satellite data from MODIS-Aqua, MODIS-Terra, VIIRS-SNPP and VIIRS-JPSS1 satellite sensors (Fig. 2).

Visual inspection of the FCA time series showed variations at multiple scales which may have different associations with relevant environmental variables, as discussed in Kahru et al. (2018). We therefore separated the time series into the longer term (called decadal) and the year-to-year or inter-annual changes. The low-frequency part of the time series ($FCA_{dec}$) was created by using a 3-year running mean on FCA. The respective high-frequency part ($FCA_{inter}$) was created by subtracting the 3-year running mean, i.e. $FCA_{inter} = FCA - FCA_{dec}$. The numerical values of the time series of FCA and its components are provided in the Supplement.

**2.2. Environmental variables**

A total of 35 time series of various environmental variables (Table 1) were compiled from both satellite-derived and in situ datasets for evaluation for relationships with estimates of cyanobacteria abundance. Details on those variables are given in the Supplement.

**2.2.1. Partial least squares (PLS) regressions**

Statistical relationships between time-series of FCA and 35 environmental variables (Table 1) were evaluated using PLS regressions as implemented in the NMath numerical libraries (http://www.centerspace.net/nmath). The analysis was started with correlations between FCA and each of the environmental variables. Combinations of an increasing number of variables were then used in PLS and the combination with the highest coefficient of determination ($R^2$) and the lowest root mean square error of prediction (RMSEP) were determined. The number of variables in a combination was increased (up to four) until no significant improvement was achieved by adding more input variables.

**3. Results**

At the decadal time scale, i.e. after smoothing with the 3-year running mean filter, the FCA time series is most strongly correlated with a group of biogeochemical variables (Fig. 3). In particular, variables related to basin-scale phosphorus availability such as DIP, P_excess, HA and N/P (see Table 1 for definitions) have the highest correlations ($0.42 < R^2 < 0.60$). The next group in the influence hierarchy on decadal scale variations in FCA is a group of *in situ* ($0-15$ m) water temperature variables (e.g. TNBP_0702-0904) with correlations exceeding the correlations with satellite-derived sea-surface temperatures. Correlations with in-water temperatures are strongest with water temperature in the Northern Baltic Proper and with water temperature at station BY15 in Eastern Gotland basin, and weaker with the water temperature averaged over the whole Baltic Proper. Variables related to solar irradiance (SDU, SID, SIS), wind speed components and phosphate concentrations in the surface layer in summer have correlation below the 95% significance at the decadal time scale.

Correlations with the inter-annual component of FCA time series, i.e. after subtracting the smoothed time series, are very different from those with the low-frequency component of FCA (Fig. 3). In general, correlations are lower than the correlations with biogeochemical variables at the decadal scale and are statistically significant only for a few variables. The strongest correlations at the inter-annual time scale are with the direct solar shortwave flux in July (two closely related variables SidAnomJul and SIDmmJul), followed by satellite-derived sea-surface temperature (SST), also in July (SSTjul). Correlations between $FCA_{inter}$ and wind components are low (near 90 % significance level) and correlations with biogeochemical variables are all practically zero.

Table 2 lists the environmental variables and combinations of variables that have the highest $R^2$ and lowest RMSEP for both $FCA_{dec}$ and $FCA_{inter}$. As environmental variables within groups are highly correlated, the next most significant variable in a combination of variables is typically from another group. It appears that the first variable provides most of the predictive power and the effect of adding additional variables is quite limited. This is particularly true for $FCA_{dec}$ for which $R^2$ increased only from 0.60 to 0.69 when increasing the number of predictor variables from one to four. For $FCA_{inter}$ the effect of adding variables was stronger, with $R^2$ increased from 0.31 to 0.45 from one variable to a combination of four variables. While the strongest correlations with individual variables at the decadal time scale are all with the biogeochemical variables (DIP, P_excess, HA and N/P), those variables are all closely correlated with each other and therefore only DIP is present in the best multi-variable combinations (Table 2). DIP alone explains 60 % of $FCA_{dec}$. The sets of variables with the highest correlations are completely different at the decadal and at the inter-annual time scales. While the in situ water temperature variables in Northern Baltic Proper (e.g. TNBP_0702-0904)) are significantly correlated with $FCA_{dec}$, they don't show up in the two-variable combinations with the highest $R^2$. For $FCA_{inter}$ the second most important variable in a combination is the July sea-surface temperature (SSTjul). The best multi-variable combinations for predicting $FCA_{dec}$ include also anomalies of the direct shortwave solar flux in August and May (SidAnomAug and SidAnomMay) but their effect is small. The respective best multi-variable combination for predicting $FCA_{inter}$ includes (in that order) the anomaly of direct solar flux in July (SidAnomJul), SST in

[Figure]

**Fig. 1.** Map of the study area. Central Baltic Proper (CBP, blue area) is the area excluding shallow coastal areas (< 15 m) and the gulfs. The sub-basins of Northern Baltic Proper (NBP), Western Gotland basin (WGB) and Eastern Gotland Basin (EGB) are shown. The small circle shows the location of the Baltic monitoring station BY15. (For interpretation of the references to colour in this figure legend, the reader is referred to the web version of this article).

July, the surface layer phosphate concentration in June-July (PO4_15m_0609-0709) and the sum of temperatures with SST over 17 °C. No variable from the in situ temperature variable group or from the basin-scale biogeochemical group is included in the best combinations for predicting $FCA_{inter}$. However, even the best 4-variable combination explains only 45 % of the variance of $FCA_{inter}$ compared to 69 % of the variance in $FCA_{dec}$ (Table 2, Fig. 4). Surprisingly, the correlations of sunshine duration averaged over July and August (SDUjul-aug) and those of the wind components are not significant (p = 0.05) when evaluated individually against $FCA_{inter}$ and are not in the best multi-variable combination for predicting $FCA_{inter}$ (Fig. 2 and Table 2). Phosphate concentrations in the surface layer in summer have no significant correlations with either $FCA_{dec}$ or $FCA_{inter}$ when taken individually but PO4_0-15m_0609-0709 is included in the best 3- and 4-variable combinations predicting $FCA_{inter}$.

**4. Discussion and conclusions**

Our analysis of the FCA time series in the central Baltic Proper (Fig. 2) showed that the environmental variables with significant correlations with FCA were almost completely different for the decadal scale and for the inter-annual scale variability in FCA. On the decadal scale, the highest correlations were clearly with basin-scale

biogeochemical variables related to pools of phosphorus and the extent of hypoxic areas. Increase of hypoxia in the Baltic Sea has been linked to increased inputs of nutrients from land (Carstensen et al., 2014). Hypoxia in bottom waters is also affected by the inflows of high-salinity North Sea waters that trigger changes in the whole water column and may lead to surface cyanobacteria blooms (Kahru et al., 2000). While the linkage between decadal scale changes in FCA and the selected biogeochemical variables makes sense as phosphorus and not nitrogen is the main limiting nutrient for $N_2$-fixing cyanobacteria, correlations do not mean causation and these correlations may be caused by similar long-term dynamics due to different reasons. However, these positive correlations support the "vicious circle" hypothesis coupling cyanobacteria blooms to anoxic conditions (Vahtera et al., 2007; Funkey et al., 2014; Savchuk, 2018). Therefore, quantitative estimation of the intensity of cyanobacteria blooms in the coming summer based on monitoring and/or modelling of the bottom water anoxic conditions during the previous winter (Janssen et al., 2004; Vahtera et al., 2007) seemed to be a promising lead towards quantitative prediction of the Baltic Sea environment. However, our results show that at the inter-annual scale, the biogeochemical variables have no influence on the variations of FCA and that makes this prospect rather dubious. Either we can hypothesize that the accuracy of the annual estimates of these biogeochemical variables is not sufficient (i.e. the inter-annual

[Figure]

**Fig. 2.** Time series (1979–2018) of the 2 components of the frequency of cyanobacteria accumulations: the low-frequency (decadal) part $FCA_{dec}$ (black filled circles, thick line, left axis) and the high-frequency (interannual) part $FCA_{inter}$ (thin line, right axis) averaged over the Central Baltic Proper (blue area in Fig. 1). (For interpretation of the references to colour in this figure legend, the reader is referred to the web version of this article).

**Table 1**

Environmental variables used in PLS to predict the low-frquency component $FCA_{dec}$ and the inter-annual component $FCA_{inter}$. $R^2$ values that are significant at $p < 0.01$ are in bold.

| Variable | Explanation | $R^2$ with $FCA_{dec}$ | $R^2$ with $FCA_{inter}$ |
|---|---|---|---|
| SSTjun | Average satellite SST for June | 0.19 | 0.01 |
| SSTjul | Average satellite SST for July | 0.07 | **0.28** |
| SSTaug | Average satellite SST for August | 0.18 | 0.04 |
| SSTjul-aug | Average satellite SST for July-August | 0.15 | 0.19 |
| DaysAbove14 | Number of days with SST above 14 °C | 0.17 | 0.07 |
| SumAbove14 | Sum of daily SSTs above 14 °C | 0.15 | 0.17 |
| DaysAbove17 | Number of days with SST above 17 °C | 0.11 | 0.15 |
| SumAbove17 | Sum of daily SSTs above 17 °C | 0.12 | 0.12 |
| TBY15_Annual | In situ annual mean temperature for $0-15$ m at BY-15 | **0.30** | 0.01 |
| TBY15_0525-0908 | Mean temperature for $0-15$ m for Jun 24 - Sep 8 at BY-15 | **0.24** | 0.06 |
| TBY15_0709-0824 | Mean temperature for $0-15$ m for Jul 9 - Aug 24 at BY-15 | **0.27** | 0.17 |
| TNBP_Annual | In situ annual mean temperature for NBP | **0.29** | 0.01 |
| TNBP_0531-0904 | In situ mean temperature for May 31 - Sep 4 for NBP | **0.33** | 0.03 |
| TNBP_0702-0904 | In situ mean temperature for Jul 2 - Sep 4 for NBP | **0.20** | 0.03 |
| TBP_Annual | In situ annual mean temperature for Baltic Proper | **0.21** | 0.02 |
| TBP_0531-0904 | In situ mean temperature for May 31 - Sep 4 for Baltic Proper | **0.22** | 0.06 |
| TBP_0702-0904 | In situ mean temperature for Jul 2 - Sep 4 for Baltic Proper | 0.17 | 0.10 |
| SDUjul-aug | Sunshine duration averaged for July-August | 0.01 | 0.09 |
| SIDmmJul | Shortwave direct irradiance averaged for July | 0.00 | **0.31** |
| SIDmmJulAug | Shortwave direct irradiance averaged for July-August | 0.00 | 0.14 |
| SidAnomMay | Shortwave direct irradiance anomaly for May | 0.08 | 0.02 |
| SidAnomJun | Shortwave direct irradiance anomaly for June | 0.09 | 0.09 |
| SidAnomJul | Shortwave direct irradiance anomaly for July | 0.00 | **0.31** |
| SidAnomAug | Shortwave direct irradiance anomaly for August | 0.01 | 0.01 |
| SidAnomJulAver3 | Shortwave direct irradiance 3 month running mean for July | 0.04 | **0.22** |
| SISmmJulAug | Shortwave irradiance averaged for July-August | 0.00 | 0.12 |
| uSeaWindsJulAug | Average eastward wind for July-August (m s$^{-1}$) | 0.00 | 0.07 |
| vSeaWindsJulAug | Average northward wind for July-August (m s$^{-1}$) | 0.03 | 0.09 |
| HA | Hypoxic area, km$^2$ | **0.53** | 0.00 |
| DIN | Dissolved inorganic nitrogen in Kt | 0.19 | 0.01 |
| DIP | Dissolved inorganic phosphorus in Kt | **0.60** | 0.01 |
| N/P | Ratio DIN/DIP | **0.42** | 0.00 |
| P_excess | Phosphorus excess in Kt | **0.55** | 0.00 |
| PO4_0525-0609 | Phosphate concentration $0-15$ m, May 25- June 9 at BY-15 | 0.00 | 0.00 |
| PO4_0609-0709 | Phosphate concentration $0-15$ m, June 9-July 9 at BY-15 | 0.00 | 0.00 |

[Figure]

**Fig. 3.** Influence of various environmental variables (coefficient of determination, $R^2$) on the low-frequency („decadal") part of FCA ($FCA_{dec}$, filled circles) and and the high-frequency („interannual") part ($FCA_{inter}$, open circles) for Central Baltic Proper. The 90 % (red dotted line), 95 % (red dashed line) and 99 % (red solid line) confidence thresholds are shown. Variables are grouped into satellite SST, In situ temperature, Solar, and biogeochemical (BGC). (For interpretation of the references to colour in this figure legend, the reader is referred to the web version of this article).

variations of these biogeochemical variables are mainly "noise") or that they truly do not matter once certain levels are reached.

Correlations with the year-to-year fluctuations are more likely to reveal causal relationships as they are less likely to be caused by co-incidental changes. However, they too are affected by random fluctuations and measurement errors in all variables. It makes sense that at the inter-annual scale (i.e. shorter scale) the variables with the highest correlations are the monthly shortwave solar flux in July and SST in July. It appears also that July is the critical month and the correlations with a number of variables in the months of May, June and August are much lower than those for July because the growth maximum occurs before the biomass peak as described by Wasmund et al. (2005). On the other hand, it is also clear that as the decadal scale changes in solar flux are minor compared to the respective changes in other variables, the

solar variables cannot be expected to be the primary variables affecting FCA at the decadal scale.

While the decadal scale changes in FCA are positively related to both in-water temperatures and satellite-detected SST, correlations with in-water temperatures are higher. This can be explained by the satellite-derived SST being affected by surface microlayer effects, making it less representative of the bulk water temperature that matters most to the cyanobacteria, even though satellite SST has superior coverage and sampling frequency. The in-water temperature at station BY15 has a particularly strong effect on indices of cyanobacteria blooms, probably due to its relatively high temporal sampling frequency. Satellite SST estimates can be affected by the surface microlayer effects at low-wind conditions in the summer (the so-called "hot spots") and near-surface blooms may actually enhance the surface water temperature (Kahru

**Table 2**

PLS predictions of FCA low-frequency (FCA$_{dec}$) and inter-annual (FCA$_{inter}$) components for the Central Baltic Proper (CBP), 1987-2018. N is the number of variables included in the prediction, $R^2$ is the coefficient of determination and RMSEP is the root mean square error of prediction.

| Predicted variable | N | Input variables | $R^2$ | RMSEP |
|---|---|---|---|---|
| FCA$_{dec}$ for Central Baltic Proper | 1 | DIP | 0.60 | 0.151 |
| | 2 | DIP, SidAnomMay | 0.62 | 0.146 |
| | 3 | DIP, SidAnomMay, SidAnomAug | 0.66 | 0.139 |
| | 4 | DIP, SidAnomMay, SidAnomAug, TNBP_0702-0904 | 0.69 | 0.132 |
| FCA$_{inter}$ for Central Baltic Proper | 1 | SidAnomJul | 0.31 | 0.194 |
| | 2 | SidAnomJul, SSTjul | 0.39 | 0.182 |
| | 3 | SidAnomJul, SSTjul, PO4_0-15m_0609-0709 | 0.43 | 0.177 |
| | 4 | SidAnomJul, SSTjul, PO4_0-15m_0609-0709, SumAbove17 | 0.45 | 0.173 |

et al., 1993). The correlations of both FCA$_{dec}$ and FCA$_{inter}$ with satellite-detected eastward and northward wind components are not significant which suggests that temporal integration of these highly variable measurements into monthly mean values may not be meaningful. It is a common perception that surface accumulations of cyanobacteria are dissipated by wind action. However, our algorithm for detecting cyanobacterial accumulations reflects not only the surface floating scum but also the backscatter from the near-surface layer just below, reducing the effect of winds on FCA.

In the northern Baltic Proper (areas NBP, WGB, EGB in Fig. 1), the year-to-year fluctuations of FCA have a quasi-regular oscillation with a period of ~3 years (Kahru et al., 2018). This seems like an internal oscillation of unknown origin and the causal links of the oscillation to environmental variables are not known. The oscillations reduce the correlations between FCA and the environmental variables included in this study. The oscillations are most evident in the northern Baltic Proper where none of the environmental variables had a $R^2$ significant at p < 0.05 level with year-to-year fluctuations in FCA (Kahru et al., 2018). When the whole central Baltic is considered, the effect of the oscillations is reduced and significant correlations are found with some

environmental variables (e.g. direct solar flux and SST in July). However, because of the variance introduced by the unexplained oscillations, environmental variables can explain only ~45 % of the total variations of FCA at inter-annual scale. The remaining ~55 % is the unexplained variance due to the internal oscillation and random errors. Comparison of FCA estimates using different satellite sensors (Kahru and Elmgren, 2014) shows that the measurement error in 2-month FCA estimates is relatively low, less than 5 %. This suggests that about half of the inter-annual variations in FCA in the central Baltic Sea cannot be estimated from environmental conditions and is probably due to biological interactions.

We hope that this work will help to improve current models of cyanobacteria dynamics (e.g. Hense, 2007) and to provide better hindcasts of the FCA time series using measured and modelled environmental variables. According to our results, in order to predict the inter-annual changes in FCA the forward models would need to predict the weather variables in July which is not going to be easy.

**Declaration of Competing Interest**

None

**Acknowledgements**

Financial support was provided by the Stockholm University's Baltic Sea Centre and its Baltic Ecosystem Adaptive Management Program and the Leibniz-Institut für Ostseeforschung Warnemünde (IOW). MK was also supported by Hanse-Wissenschaftskolleg (Delmenhorst, Germany) and OPS by the Swedish Agency for Marine and Water Management through the Baltic Nest Institute with their Grant 1:11. We thank NASA Ocean Color Processing Group (OBPG) and CM SAF for satellite data. [CG]

**Appendix A. Supplementary data**

Supplementary material related to this article can be found, in the online version, at doi:https://doi.org/10.1016/j.hal.2019.101739.

[Figure]

[Figure]

**Fig. 4.** Predicting the frequency of cyanobacteria accumulations (FCA) in the central Baltic Sea from time series of environmental variables with partial least squares (PLS) regression. The time series have been normalized to a range [0, 1], the observed series is shown with a blue line and the predicted series with a red dashed line with open diamonds. A, FCA low-frequency component (FCA$_{dec}$) predicted from DIP, SidAnomMay, SidAnomAug, TNBP_0702-0904 ($R^2$ = 0.69); B, FCA inter-annual component (FCA$_{inter}$) predicted from SidAnomJul, SSTjul, PO4_0-15m_0609-0709, SumAbove17 ($R^2$ = 0.45). The variable names are listed in Table 1. (For interpretation of the references to colour in this figure legend, the reader is referred to the web version of this article).

**References**

Bianchi, T.S., Engelhaupt, E., Westman, P., Andrén, T., Rolff, C., Elmgren, R., 2000. Cyanobacterial blooms in the Baltic Sea: natural or human-induced? Limnol. Oceanogr. 45, 716–726. https://doi.org/10.4319/lo.2000.45.3.0716.

Carstensen, J., Andersen, J.H., Gustafsson, B.G., Conley, D.J., 2014. Deoxygenation of the Baltic Sea during the last century. Proc. Nat. Acad. Sci. USA 111, 5628–5633.

Center Space Software, 2016. NMath Stats User's Guide, Version 4.2. Center Space Software, Corvallis, OR. http://www.centerspace.net/.

Funkey, C.P., Conley, D.J., Reuss, N.S., Humborg, C., Jilbert, T., Slomp, C.P., 2014. Hypoxia sustains cyanobacteria blooms in the Baltic Sea. Environ. Sci. Technol. 48, 2598–2602.

Hense, I., 2007. Regulative feedback mechanisms in cyanobacteria-driven systems: a model study. Mar. Ecol. Prog. Ser. 339, 41–47.

Hajdu, S., Höglander, H., Larsson, U., 2007. Phytoplankton vertical distributions and composition in Baltic Sea cyanobacterial blooms. Harmful Algae 6, 187–205.

Janssen, F., Neumann, T., Schmidt, M., 2004. Inter-annual variability in cyanobacteria blooms in the Baltic Sea controlled by wintertime hydrographic conditions. Mar. Ecol. Prog. Ser. 275, 59–68.

Karlson, A.M.L., Duberg, J., Motwani, N.H., Hogfors, H., Klawonn, I., Ploug, H., Svedén, J.B., Garbaras, A., Sundelin, B., Hajdu, S., Larsson, U., Elmgren, R., Gorokhova, E., 2015. Nitrogen fixation by cyanobacteria stimulates production in Baltic food-webs. Ambio 44, 413–426.

Kahru, M., Leppänen, J.-M., Rud, O., 1993. Cyanobacterial blooms cause heating of the sea surface. Mar. Ecol. Prog. Ser. 101, 1–7.

Kahru, M., Leppänen, J.-M., Rud, O., Savchuk, O.P., 2000. Cyanobacteria blooms in the Gulf of Finland triggered by saltwater inflow into the Baltic Sea. Mar. Ecol. Prog. Ser. 207, 13–18.

Kahru, M., Horstmann, U., Rud, O., 1994. Satellite detection of increased cyanobacteria blooms in the Baltic Sea: Natural fluctuation or ecosystem change? Ambio 23, 469–472.

Kahru, M., Savchuk, O.P., Elmgren, R., 2007. Satellite measurements of cyanobacterial bloom frequency in the Baltic Sea: interannual and spatial variability. Mar. Ecol.-Prog. Ser. 343, 15–23.

Kahru, M., Elmgren, R., 2014. Multidecadal time series of satellite-detected accumulations of cyanobacteria in the Baltic Sea. Biogeosciences 11, 3619–3633. https://doi.org/10.5194/bg-11-3619-2014.

Kahru, M., Elmgren, R., Savchuk, O.P., 2018. Unexplained interannual oscillations of cyanobacterial blooms in the Baltic Sea. Sci. Rep. 8, 6365. https://doi.org/10.1038/s41598-018-24829-7.

Kutser, T., 2004. Quantitative detection of chlorophyll in cyanobacterial blooms by satellite remote sensing. Limnol. Oceanogr. 49, 2179–2189.

Larsson, U., et al., 2001. Baltic Sea nitrogen fixation estimated from the summer increase in upper mixed layer total nitrogen. Limnol. Oceanogr. 46, 811–820.

Niemi, Å., 1979. Blue-green algal blooms and N:P ratio in the Baltic Sea. Acta Bot. Fenn. 110, 57–61.

Paerl, H.W., Huisman, J., 2008. Blooms like it hot. Science 320, 57–58.

Paerl, H.W., Huisman, J., 2009. Climate change: a catalyst for global expansion of harmful cyanobacterial blooms. Environ. Microbiol. 1, 27–37.

Paerl, H.W., Xu, H., McCarthy, M.J., Zhu, G., Quin, B., Yi, L., Gardner, W.S., 2011. Controlling harmful cyanobacterial blooms in a hyper-eutrophic lake (Lake Taihu, China): the need for a dual nutrient (N and P) management strategy. Water Res. 45 (5), 1973–1983.

Rolff, C., Almesjö, L., Elmgren, R., 2007. Nitrogen fixation and the abundance of the diazotrophic cyanobacterium *Aphanizomenon* sp. in the Baltic Proper. Mar. Ecol. Prog. Ser. 332, 107–118.

Savchuk, O.P., 2018. Large-scale nutrient dynamics in the Baltic Sea, 1970–2016. Front. Mar. Sci. 5. https://doi.org/10.3389/fmars.2018.00095.

Vahtera, E., Conley, D.J., Gustafsson, B.G., Kuosa, H., Pitkanen, H., Savchuk, O.P., Tamminen, T., Viitasalo, M., Voss, M., Wasmund, N., Wulff, F., 2007. Internal ecosystem feedbacks enhance nitrogen-fixing cyanobacteria blooms and complicate management in the Baltic Sea. Ambio 36, 186–194.

Wasmund, N., 1997. Occurrence of cyanobacterial blooms in the Baltic Sea in relation to environmental conditions. Int. Revue Gesamten Hydrobiol. 82, 169–184.

Wasmund, N., Nausch, G., Schneider, B., Nagel, K., Voss, M., 2005. Comparison of nitrogen fixation rates determined with different methods: a study in the Baltic Proper. Mar. Ecol. Prog. Ser. 297, 23–31.